# Long-term evaluation of surface air pollution in CAMSRA and MERRA-2 global reanalyses over Europe (2003-2020)

Aleks Lacima[1], Hervé Petetin[1], Albert Soret[1], Dene Bowdalo[1], Oriol Jorba[1], Zhaoyue Chen[2,3], Raúl Méndez Turrubiates[2], Hicham Achebak[2], Joan Ballester[2], and Carlos Pérez García-Pando[1,4]

[1]Earth Sciences Department, Barcelona Supercomputing Center, Barcelona, Spain
[2]ISGlobal, Barcelona, Spain
[3]Universitat Pompeu Fabra (UPF), Barcelona, Spain
[4]ICREA, Catalan Institution for Research and Advanced Studies, Barcelona, Spain

**Correspondence:** Aleksander Lacima (aleksander.lacima@bsc.es)

**Abstract.** Over the last century, our societies have experienced a sharp increase in urban population and fossil-fueled transportation, turning air pollution into a critical issue. It is therefore key to accurately characterize the spatiotemporal variability of surface air pollution, in order to understand its effects upon the environment, knowledge that can then be used to design effective pollution reduction policies. Global atmospheric composition reanalyses offer great capabilities towards this char-
5 acterization through assimilation of satellite measurements. However, they generally do not integrate surface measurements and thus remain affected by significant biases at ground-level. In this study, we thoroughly evaluate two global atmospheric composition reanalyses, CAMSRA and MERRA-2, between 2003 and 2020, against independent surface measurements of $O_3$, $NO_2$, CO, $SO_2$, $PM_{10}$ and $PM_{2.5}$ over the European continent. Overall, both reanalyses present significant and persistent biases for almost all examined pollutants. CAMSRA clearly outperforms MERRA-2 in capturing the spatiotemporal variabil-
10 ity of most pollutants, as shown by generally lower biases (all pollutants except for $PM_{2.5}$), lower errors (all pollutants) and higher correlations (all pollutants except $SO_2$). CAMSRA also outperforms MERRA-2 in capturing the annual trends found in all pollutants (except for $SO_2$). Overall, CAMSRA tends to perform best for $O_3$ and CO, followed by $NO_2$ and $PM_{10}$, while poorer results are typically found for $SO_2$ and $PM_{2.5}$. Higher correlations are generally found in autumn/winter for reactive gases. Compared to MERRA-2, CAMSRA assimilates a wider range of satellite products which, while enhancing the perfor-
15 mance of the reanalysis in the troposphere (as shown by other studies), has a limited impact on the surface. The biases found in both reanalyses are likely explained by a combination of factors, including errors in emission inventories and/or sinks, a lack of surface data assimilation and their relatively coarse resolution. Our results highlight the current limitations of reanalyses to represent surface pollution, which limits their applicability for health and environmental impact studies. When applied to reanalysis data, bias-correction methodologies based on surface observations should help to constrain the spatiotemporal
variability of surface pollution and its associated impacts.

# 1 Introduction

In the last two decades, reanalyses have become a very powerful tool in modern Earth sciences as they combine both model- and observation-based information to provide physically consistent data of land, ocean and atmospheric variables with continuous spatial and temporal coverage. In the field of atmospheric composition, different reanalysis products are available at global scale, including the Copernicus Atmospheric Monitoring Service reanalysis (CAMSRA; Inness et al. (2019)), produced by the European Centre for Medium-Range Weather Forecasts (ECMWF), and the Modern-Era Retrospective Analysis for Research and Applications v2 (MERRA-2; Gelaro et al. (2017), Randles et al. (2017), Buchard et al. (2017a)), produced by National Aeronautics and Space Administration (NASA)'s Global Modelling and Assimilation Office (GMAO). Both products assimilate a variety of space-based remote sensing observations (mostly total and tropospheric columns) obtained from a growing fleet of satellites measuring reactive gases such as ozone ($O_3$), nitrogen dioxide ($NO_2$) or carbon monoxide (CO), as well as aerosol optical depth (AOD). Such an extensive data assimilation of satellite observations is crucial for reducing the biases related to erroneous emission forcings and/or overly coarse representations of the physical and chemical processes that occur in the atmosphere. Data assimilation helps better constrain the spatiotemporal variability and long-term trends of the most important chemical compounds, providing a physically consistent view of the Earth's atmospheric composition.

Considering the strong interest of atmospheric composition reanalyses for a variety of applications (e.g. climatological studies, initial and/or boundary conditions for regional-scale modeling systems, air pollution impact assessment and health studies), it is crucial to characterize the strengths and limitations of these global products, in particular at the surface, as no in situ chemical observations are assimilated. The most recent studies evaluating the CAMSRA and/or MERRA-2 reanalysis at ground-level are indicated in Table 1, highlighting the limited effort that has been made so far to evaluate and inter-compare these reanalysis products against in situ surface measurements.

The main findings of this more recent literature are briefly outlined here. Ryu and Min (2021) found significant and persistent biases in all the pollutants examined over South Korea, with CAMSRA outperforming MERRA-2 in all cases except for $SO_2$. At global scale, Wagner et al. (2021) showed that CAMSRA provides an overall accurate representation of reactive gases over time, and highlighted the key role played by satellite data assimilation in improving atmospheric composition reanalysis products. Both these two previous studies analyze a wide range of aerosols and reactive gases and cover the most extensive period possible at the time, 2003-2018, which is limited by the start of CAMSRA in 2003. Ma et al. (2021) found persistent negative biases in $PM_{10}$ concentration over mainland China in MERRA-2 for the periods 2011-2013 and 2016-2017, with better performance during summer. Their results also showed a significant improvement when including nitrate compounds. Navinya et al. (2020) found a systematic underestimation of $PM_{2.5}$ concentration in MERRA-2 over India for the period 2015-2018. Huijnen et al. (2020) found limited surface $O_3$ biases when evaluating CAMSRA over Europe (-1.8 ppbv). Ukhov et al. (2020) evaluated surface $SO_2$ for 2015-2016 over three cities in the Middle East and found a large underestimation for MERRA-2, while CAMSRA showed both moderate negative and positive biases. Lastly, Ali et al. (2022) evaluated PM over the period 2014-2020 in China and found significant over- and underestimations both for CAMSRA and MERRA-2.

**Table 1.** Review of recent studies evaluating the CAMSRA and/or MERRA-2 reanalysis at the surface using in situ observations.

| Author | Region | Period | Reanalysis | Pollutants |
|---|---|---|---|---|
| Ryu and Min (2021) | South Korea | 2003-2018 | CAMSRA; MERRA-2; TCR-2 | $CO$, $NO_2$, $SO_2$, $O_3$, $PM_{10}$ |
| Wagner et al. (2021) | Global | 2003-2018 | CAMSRA | $NO_2$, $O_3$, $CO$, $HCHO$ |
| Ma et al. (2021) | China | 2011-2013; 2016-2017 | MERRA-2 | $PM_{10}$ |
| Navinya et al. (2020) | India | 2015-2018 | MERRA-2 | $PM_{2.5}$ |
| Provençal et al. (2017a) | Europe | 2003-2014 | MERRA-1 | $PM_{2.5}$, $PM_{10}$ |
| Provençal et al. (2017b) | Israel; Taiwan | 2002-2015 | MERRA-1 | $PM_{2.5}$ |
| Buchard et al. (2016) | USA | 2003-2012 | MERRA-1 | $PM_{2.5}$ |
| Huijnen et al. (2020) | Global | 2003-2015 | CAMSRA; TCR | $O_3$ |
| Ukhov et al. (2020) | Middle East | 2015-2016 | CAMSRA; MERRA-2 | $SO_2$ |
| Ali et al. (2022) | China | 2014-2020 | CAMSRA; MERRA-2 | $PM_{10}$, $PM_{2.5}$ |

Our study evaluates CAMSRA and MERRA-2 against independent surface in situ measurements over the period 2003-2020, focusing on the European continent, a region still poorly covered by past evaluation studies (Table 1). It considers all major pollutants with recognized harmful effects on human health and sufficient observational data available at the surface, namely $O_3$, $NO_2$, $CO$, $SO_2$, $PM_{10}$ and $PM_{2.5}$. The motivation behind this study arose in the context of the European Research Council (ERC) project EARLY-ADAPT (https://early-adapt.eu/), in the frame of which a pioneer health dataset is currently being collected over Europe to investigate the time-varying health effects of climate and air pollution, and thus shed light into the early adaptation response to climate change in the field of human health. This impact will be quantified by fitting epidemiological models on historical local health, climate and air pollution data, which thus requires a long-term (multi-decadal) air quality database of the most harmful pollutants, at daily-scale and over the entire European domain. Despite their relatively coarse spatial resolution, which is the counterpart to a sufficiently long-term coverage, global-scale atmospheric composition reanalyses provide highly valuable information, though remain subject to biases and errors both in terms of spatial, seasonal and intra-annual variability, but also regarding long-term trends. It is worth mentioning here that the CAMS regional reanalysis (Marécal et al. (2015)), focused on Europe, assimilates surface in situ observations and provides air pollution fields at a finer spatial resolution than CAMSRA, but only over a limited period of time (2014-2018), for which reason we focus here on the global reanalysis.

In Sect. 2, we introduce the data (Subsect. 2.1) and provide details on the different methods employed for their analysis (Subsect. 2.2). Results are presented and discussed in Sect. 3, and summarized in Sect. 4.

## 2 Data & Methodology

In this section we briefly describe our observational and reanalysis datasets, while providing details on the different statistical methods employed for their analysis. Throughout this work, square brackets, [], are used to indicate concentration or mixing

ratio of a chemical compound (e.g. $[O_3]$ = $O_3$ mixing ratio, $[PM_{10}]$ = $PM_{10}$ concentration) measured in *parts per billion*
(ppbv) for reactive gases and in $\mu g\,m^{-3}$ for aerosols. Nonetheless, the term concentration is used for the sake of simplicity
when reactive gases are mentioned together with aerosols.

## 2.1 Data

Our model data come from two global atmospheric composition reanalyses, CAMSRA and MERRA-2, whose main char-
acteristics are summarized in Table 2. The reanalyses are evaluated against surface in situ measurements obtained from two
European Environment Agency (EEA) databases, AIRBASE, for the period 2003-2012 (EEA, 2014), and AQ e-Reporting
(EEA, 2018), for the period 2012-2020. No significant inconsistencies are expected between AIRBASE and AQ e-Reporting
given that stations included in both databases are obtained from the same network. Though stations may be renamed, relocated,
or even removed with time, this is not expected to significantly affect our data given the large number of stations considered
and the continuous addition of new stations into the network throughout all the period 2003-2020.

## 2.1.1 CAMSRA

Produced by ECMWF, the CAMS global atmospheric composition reanalysis consists of 3-dimensional time-consistent at-
mospheric composition fields that include chemical species, aerosols and greenhouse gases (GHGs), and currently covers a
temporal period extending from 2003 to mid-2021. The reanalysis starts in 2003, when space-based observational measure-
ments, retrieved from a myriad of instruments on-board Envisat, Terra, Aura, MetOp and POES satellites, became available.
The latest CAMSRA version was produced in cycle 42R1 of ECMWF's Integrated Forecast System (IFS) using 4DVar data
assimilation of satellite measurements, including $O_3$, $NO_2$, CO and AOD. This IFS cycle includes the modified Carbon Bond
2005 Chemical Mechanism (CB05), which serves as the tropospheric chemistry scheme of the reanalysis (Flemming et al.,
2015). Anthropogenic emissions come from the MACCity inventory data (Granier et al., 2011) for the period 2003-2010, and
from 2010 onwards they are derived according to the representative concentration pathway of 8.5 $Wm^{-2}$ (RCP8.5). Biomass
burning emissions are obtained from the Global Fire Assimilation System (GFAS) v1.2 (Kaiser et al., 2012), whereas monthly
mean biogenic volatile organic compound (VOC) emissions are computed with the Model of Emissions of Gases and Aerosols
from Nature (MEGAN) using MERRA-2 reanalysed meteorology (Sindelarova et al., 2014). Meteorological observations are
assimilated as in ERA5 (Hersbach et al., 2020).

CAMSRA has a horizontal resolution of approximately 80 km (similar to a regular 0.75º x 0.75º latitude/longitude grid),
with atmospheric composition fields being available only in grid-point space. Its vertical resolution consists of 60 hybrid
sigma/pressure model levels, with the top of the first level at 10 m above ground and the top level located at 0.1 hPa. CAMSRA
products are available at a temporal resolution of 3 h, including 3-hourly analysis fields and 3-hourly forecast fields. The bi-
ases present in the different AC satellite-retrieved datasets employed to build CAMSRA is corrected through a variational bias
correction scheme (Dee and Uppala, 2008). For a more thorough and detailed description of CAMSRA we direct the reader to
Inness et al. (2019) and Wagner et al. (2021).

In CAMSRA, both $PM_{10}$ and $PM_{2.5}$ are directly available and do not require to be reconstructed from its separate aerosol

compounds, which include black carbon (BC), organic carbon (OC), organic matter (OM), sulphate ($SO_4$), sea salt and dust. Both PM fields were downloaded directly without any reconstruction or modification, though they are originally reconstructed from the following formulas:

$$[\mathrm{PM}_{10}] = \rho \left( \frac{[\mathrm{SS1}]}{4.3} + \frac{[\mathrm{SS2}]}{4.3} + [\mathrm{DD1}] + [\mathrm{DD2}] + 0.4[\mathrm{DD3}] + [\mathrm{OM1}] + [\mathrm{OM2}] + [\mathrm{SU1}] + [\mathrm{BC1}] + [\mathrm{BC2}] \right) \tag{1a}$$

$$[\mathrm{PM}_{2.5}] = \rho \left( \frac{[\mathrm{SS1}]}{4.3} + 0.5\frac{[\mathrm{SS2}]}{4.3} + [\mathrm{DD1}] + [\mathrm{DD2}] + 0.4[\mathrm{DD3}] + 0.7[\mathrm{OM1}] + 0.7[\mathrm{OM2}] + 0.7[\mathrm{SU1}] + [\mathrm{BC1}] + [\mathrm{BC2}] \right)$$
$$\tag{1b}$$

Where $\rho$ is the air density, SS1/2 the sea salt, DD1/2/3 the dust, OM1/2 the organic matter, BC1/2 the black carbon, and SU1 the aerosol sulfate mass mixing ratios (with 1/2/3 referring to the aerosol bins, from smallest to largest). The factor 4.3 is applied to convert the model sea salts, expressed at 80 % relative humidity in the model (see Reddy et al. (2005)), into dry mass mixing ratios. However, it is worth mentioning that to the best of our knowledge, this correction might need to be revisited in the future to also account for the change of size of the sea salt particles (as mentioned on the CAMS scientific user forum: https://confluence.ecmwf.int/display/CUSF/PM10+and+PM25+global+products, last access: 25th November, 2022). Notably, aerosol nitrates are, at this time, not included in the reanalysis, which could in principle lead to significant underestimations in regions where nitrates represent an important part of total aerosol concentration. Although in practice, the assimilation of AOD observations (that evidently integrate all the aerosol compounds) is expected to reduce these biases. Within OM, secondary organic aerosols (SOA) of anthropogenic origin are parametrized according to Spracklen et al. (2011), based on MACCity CO emissions. A detailed description of the aerosol scheme employed in CAMSRA can be found in Morcrette et al. (2009).

**Table 2.** Summary of reanalysis products.

| Reanalysis | CAMSRA | MERRA-2 |
|---|---|---|
| Available pollutants | $O_3$, $NO_2$, CO, $SO_2$, $PM_{10}$, $PM_{2.5}$ | $O_3$, CO, $SO_2$, $PM_{10}$, $PM_{2.5}$ |
| Coverage period | 2003–present | 1980–present |
| Spatial resolution | ~80 km (roughly 0.75º x 0.75º) | 0.5° x 0.625° |
| Assimilation system | 4D-Var | 3D-Var Gridpoint Statistical Interpolation (GSI) |
| Meteorology | IFS Cycle 42r1 (Hersbach et al., 2020) | GEOS-5 (Rienecker et al. (2008), Molod et al. (2012)) |
| Chemistry | IFS(CB05) (Flemming et al., 2015) | GOCART (Chin et al. (2002), Colarco et al. (2010)) |
| Anthropogenic emissions | MACCity (Granier et al., 2011) | AeroCom Phase II (HCA0 v1; Diehl et al. (2012)), EDGARv4.2 (https://edgar.jrc.ec.europa.eu/) |
| Biomass burning emissions | GFASv1.2 (Kaiser et al., 2012) | RETROv2 (Duncan et al., 2003), GFEDv3.1 (Randerson et al., 2006), QFED 2.4-r6 (Darmenov and da Silva, 2013) |
| Biogenic emissions | MEGAN (Sindelarova et al., 2014) | NVOC (Guenther et al., 1995) |
| Volcanic emissions | — | AeroCom Phase II (HCA0 v2; Diehl et al. (2012)) |
| Assimilated $O_3$ products | SCIAMACHY, MIPAS, MLS OMI, GOME-2, SBUV/2 | MLS, OMI, SBUV, SBUV/2 |
| Assimilated $NO_2$ products | SCIAMACHY, OMI, GOME-2 | — |
| Assimilated CO products | MOPITT | — |
| Assimilated $SO_2$ products | — | — |
| Assimilated aerosol products | AATSR, MODIS | AVHRR, AERONET, MISR, MODIS |

### 2.1.2 MERRA-2

Developed by NASA's GMAO, the MERRA-2 atmospheric composition reanalysis is based on the Goddard Earth Observing
System v5 (GEOS-5) atmospheric model. It is important to note at this stage that, in contrast with CAMSRA, which aims
to simulate all major chemical compounds present in the atmosphere, the MERRA-2 reanalysis, despite being the first atmo-
spheric composition reanalysis that couples chemistry to global atmospheric circulation, focuses mainly on aerosols. Therefore,
aside from meteorological data, only AOD observations and $O_3$ columns are assimilated in MERRA-2, based on both measure-
ments from Terra, Aura, MetOp and POES satellites, and - unlike in CAMSRA - surface-based observations from the Aerosol
Robotic Network (AERONET). Anthropogenic sulfate, black carbon (BC) and primary organic matter (POM) emissions are
obtained from AEROsol COMparisons between Observations and Models (AeroCom) Phase II (HCA0 v1; Diehl et al. (2012)).
Anthropogenic $SO_2$ emissions are taken from the Emissions Database for Global Atmospheric Research (EDGAR) v4.2, devel-
oped by the European Commission (https://edgar.jrc.ec.europa.eu/), whereas volcanic $SO_2$ is retrieved from AeroCom Phase II
(HCA0 v2; Diehl et al. (2012)). CO is simulated by the GEOS-5 modeling system. Sea salt and dust emissions, both composed
of five non-interacting size bins, are wind-driven. Aerosol chemistry is reproduced with a version of the Goddard Chemistry

Aerosol Radiation and Transport (GOCART; Chin et al. (2002), Colarco et al. (2010)) model, which simulates the processes, interactions, sources and sinks of the different chemical compounds included in MERRA-2, with the exception of $O_3$ and CO. MERRA-2 currently covers a temporal period extending from 1980 to mid-2021. The reanalysis was produced using 3DVar data assimilation of AOD and several other meteorological fields. MERRA-2 uses cubed-sphere horizontal discretization,

which serves to mitigate grid spacing singularities that appear in regular Gaussian grids, at an approximate resolution of 0.5° x 0.625° (~50 km), and has 72 hybrid-eta model levels from the surface, with the first level reaching 58 m above ground, to the top at 0.01 hPa. MERRA-2 includes 1-hourly and 3-hourly analysis fields for its aerosol diagnostics and meteorological data. For a more thorough and detailed description of MERRA-2 we direct the reader to Gelaro et al. (2017) and Randles et al. (2017).

Designed primarily for research focused on aerosols, the MERRA-2 reanalysis dataset also provides data of the most important trace gases, including $O_3$, CO and $SO_2$ (with only $NO_2$ being unavailable). In MERRA-2, both $PM_{10}$ and $PM_{2.5}$ need to be reconstructed from the available aerosol chemical compounds, which include organic carbon (OC), black carbon (BC), dust (DS), sea salt (SS) and sulphate ($SO_4$). In this study, the $PM_{10}$ and $PM_{2.5}$ concentrations are computed as follows:

$$[PM_{10}] = 1.375[SO_4] + 1.8[OC] + [BC] + [DS] + [SS] \tag{2a}$$

$$[PM_{2.5}] = 1.375[SO_4] + 1.8[OC] + [BC] + [DS_{2.5}] + [SS_{2.5}] \tag{2b}$$

The 1.375 factor applied to $[SO_4]$ is used here to convert sulfate into ammonium sulfate (assuming full neutralization). The 1.8 factor applied to $[OC]$ accounts for other organic compounds found in organic matter (OM). In recent literature, Eq. 2a and 2b are the most frequently used to reconstruct the PM fields. Equation 2a is used by Provençal et al. (2017b) and also in Ma et al. (2021), though with an additional term to account for aerosol nitrates in the latter. Equation 2b is used by Provençal et al.

(2017a, b) and in Ryu and Min (2021), where it is also employed to reconstruct $[PM_{10}]$ by multiplying it with a measurement-based $[PM_{10}]/[PM_{2.5}]$ ratio of 1.75 (computed over the period 2003-2018). Note also that there are large uncertainties in the $[OM]/[OC]$ ratio as it varies in time and space, and other studies have chosen a different value (e.g. 1.4 in Buchard et al. (2016) and Buchard et al. (2017b)) for this factor. Notably, nitrates are currently not available in MERRA-2, even though they can make up a considerable portion of total [PM] Aldabe et al. (2011). To overcome this limitation, some authors such as Ma et al.

(2021) have introduced an additional term partly based on observations.

In our study aerosol nitrates are not included in the $PM_{10}$ and $PM_{2.5}$ concentration fields, neither in MERRA-2 nor in CAM-SRA. The potential underestimation due to the absence of nitrates is at least partially compensated by the fact that both reanalyses assimilate total AOD observations, which corrects all PM chemical compounds proportionally and thus minimizes the biases due to the absence of aerosol nitrates.

**2.1.3   Air quality observations and GHOST**

The EEA observations are accessed from the Globally Harmonised Observational Surface Treatment (GHOST) initiative, a Barcelona Supercomputing Center (BSC) in-house project dedicated to the harmonisation of global air pollution surface observations and its metadata, with the purpose of facilitating a greater quality of observational/model comparison in the atmo-

spheric chemistry community. Besides the chemical concentration data originally available in the EEA databases, GHOST provides an extended set of metadata, including a variety of quality-assurance (QA) flags, that is used here to eliminate doubtful, non-physical or other faulty data (see Appendix D for a detailed description of the QA filters applied here). To ensure a good temporal representativeness, only daily averages based on at least 18 hourly values (75% threshold) are retained in our study. Given the relatively coarse spatial resolution of both reanalyses, only rural, rural-regional and rural-remote background stations, of larger spatial representativeness, are considered in the evaluation, thus excluding urban and suburban background stations. Traffic and industrial point source stations have also been discarded, being generally located in areas with limited air flow and close to local emission sources, which causes their pollution concentration levels to be overly driven by day-to-day variability. For information purpose, evaluation results obtained considering only urban and suburban background stations will be also briefly discussed. More information on the station classification can be found on the EEA website (https://www.eea.europa.eu/themes/air/air-quality-concentrations/classification-of-monitoring-stations-and, last access: 15th December 2022).

## 2.2 Methodology

Our domain of study extends from 25°W to 45°E in longitude, and from 27°N to 72°N in latitude, thus covering all continental Europe as well as the Canary Islands, Iceland, Western/European Russia, North Africa and the westernmost regions of the Middle East and the Caucasus. For convenience, both CAMSRA and MERRA-2 are regridded over this domain on a common regular longitude-latitude grid at a resolution of 0.2º x 0.2º (roughly 20 $\mathrm{km}$) through bilinear interpolation. The (pointwise) observations are also gridded to this same resolution by averaging (at daily-scale) all the stations available within a given grid cell. Compared to a pointwise-to-gridded comparison, this is expected to partly overcome the issues of spatial representativeness and spatial heterogeneity, although we acknowledge here that more sophisticated methods such as those proposed by Souri et al. (2022) (which employ geostatistical approaches by making use of semivariograms and kriging) might be worth implementing in the future. However, when considering only rural, rural-regional and rural-remote background stations, the proportion of gridded daily observations based on one single daily observation (two daily observations) is 96.1 % (3.5 %) for $NO_2$, 95.4 % (4.4 %) for $O_3$, 96.7 % (3.2 %) for $SO_2$, 97.9 % (1.9 %) for CO, 91.0 % (8.5 %) for $PM_{10}$ and 92.5 % (7.4 %) for $PM_{2.5}$, these high percentages being explained by the presence of numerous missing values throughout the period of study. Table 3 and Fig. 1 provide some information on the observations available over our European domain during 2003-2020, in terms of both pointwise and gridded observations (the total number of observations is typically reduced by a 2-3 factor after the gridding operation). Unfortunately, in situ observations from GHOST are not available for several countries falling within the domain considered in this study, located in Northern Africa (e.g. Morocco, Algeria, Tunis, Lybia, Egypt), Eastern Europe (e.g. Russia, Belarus, Ukraine) and the Middle East (e.g. Israel, Lebanon, Jordan, Syria), thus somewhat limiting the scope of the evaluation, particularly in terms of spatial variability and pollution hot-spots.

**Table 3.** Number of EEA background stations (S), number of gridded stations (G) and number of overall points (i.e. daily values) (N) over the period 2003-2020.

| Pollutant | EEA stations | $S_{rural}$ | $G_{rural}$ | $N_{points}$ ($10^6$) | $S_{urban}$ | $G_{urban}$ | $N_{points}$ ($10^6$) |
|-----------|-------------|-------------|-------------|----------------------|-------------|-------------|----------------------|
| $O_3$ | 5701 | 1511 | 728 | 3.04 | 4190 | 1278 | 5.13 |
| $NO_2$ | 8381 | 1460 | 609 | 2.10 | 6921 | 1461 | 5.52 |
| CO | 2584 | 200 | 89 | 0.16 | 2384 | 553 | 1.13 |
| $SO_2$ | 5424 | 1050 | 443 | 0.77 | 4374 | 1147 | 2.34 |
| $PM_{10}$ | 9500 | 1475 | 542 | 1.83 | 8025 | 1566 | 5.84 |
| $PM_{2.5}$ | 3874 | 632 | 291 | 0.75 | 3242 | 907 | 2.35 |

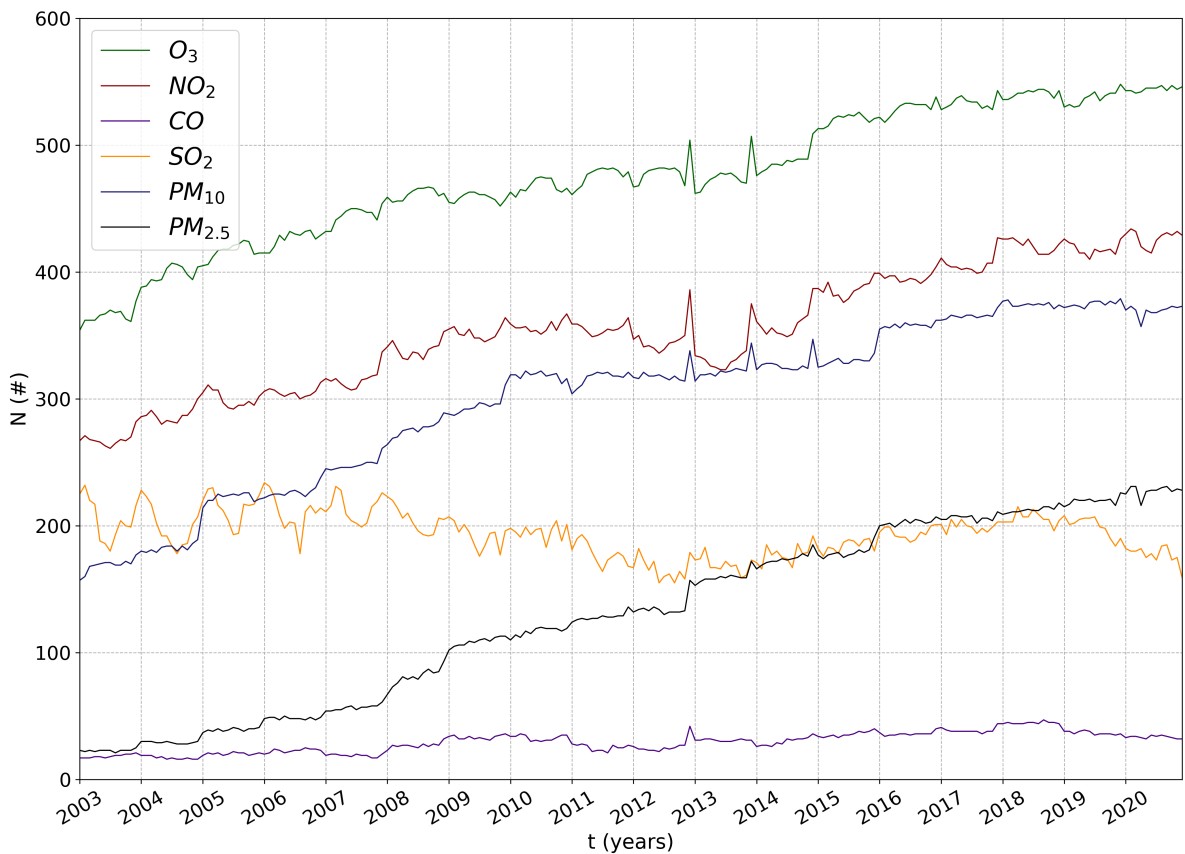

**Figure 1.** Monthly number of rural gridded cells with available observational data for $O_3$, $NO_2$, CO, $SO_2$, $PM_{10}$ and $PM_{2.5}$ over the period 2003-2020.

The evaluation is performed on a set of metrics including the (normalized) Mean Bias ((n)MB), the (normalized) Root Mean Square Error ((n)RMSE) and the Pearson Correlation Coefficient (PCC), defined as follows:

$$\text{MB} = \frac{1}{N} \sum_{i=1}^{N} (m_i - o_i) \tag{3a}$$

$$\text{nMB} = \frac{\text{MB}}{\overline{o}} \times 100\% \tag{3b}$$

$$\text{RMSE} = \sqrt{\frac{\sum_{i=1}^{N} (m_i - o_i)^2}{N}} \tag{3c}$$

$$\text{nRMSE} = \frac{\text{RMSE}}{\overline{o}} \times 100\% \tag{3d}$$

$$\text{PCC} = \frac{1}{N-1} \sum_{i=1}^{N} \frac{(m_i - \overline{m})(o_i - \overline{o})}{\sigma_m \sigma_o} \tag{3e}$$

Where $m_i$ and $o_i$ are the predicted and observed concentrations, $\overline{m}$ and $\overline{o}$ their means, $\sigma_m$ and $\sigma_o$ their standard deviations, and $N$ is the number of points employed to compute the statistics (i.e. number of daily values across all stations). The index $i$ accumulates over time (e.g. daily, monthly) at each station (i.e. gridded cell with available observations). The final value for each statistic is obtained by medianizing across all stations. The overlines in Eq. 3a–3e indicate a time-averaged variable.

In this study, metrics have been calculated and presented following two different approaches: (1) with a so-called "time-and-space" approach where metrics are calculated in one step, based on all reanalysis-observation pairs available both across the entire domain (or a given country) and over the entire period 2003-2020, or (2) with a so-called "time-then-space" approach where metrics are first calculated at each station, before being combined by taking the median across all stations. In this work-frame, "time-and-space" PCC values do not correspond to spatial or temporal correlations but rather to overall spatio-temporal correlations, while "time-then-space" PCC values do correspond to temporal correlations, though spatially averaged.

Annual trends, based on monthly averages over the entire domain (considering only cells and days with available observations to allow for fair comparisons) and reported in Sect. 3, have been computed using Seasonal Theil-Sen estimators, which account for seasonal variability. Statistical significance has been analyzed through correlated seasonal Mann-Kendall trend tests, considering both seasonality and autocorrelation. For more detailed information on how the annual trends are computed we refer the reader to Appendix C. It is worth noting that trends are here computed essentially to evaluate the consistency of the reanalyses against observational data, but should not be taken as a reliable estimate of real pollutant trends due to the number of stations not being constant, but generally increasing throughout the period of study. Moreover, even if a station has available data over the entire period, its location can also be subject to changes over time.

## 3  Results & Discussion

The evaluation results, alongside its analysis and discussion, are presented in this section. Overall statistics obtained over the European continent during 2003-2020 are provided in Table 4 ("time-and-space" approach). Annual trends are reported in Table 5 for the different pollutants.

Different aspects of the evaluation results are provided for each pollutant in Figs. 2-7, including (1) monthly time series of concentrations and evaluation statistics, (2) bar plots of country-scale statistics, and (3) maps of mean concentrations (and differences between both reanalyses) over the domain. Each point in the monthly time series corresponds to the median of the monthly-mean values across all individual cells with available observations over the domain. In order to highlight potential spatial differences in pollution patterns across the European continent, country-scale statistics computed over the entire time period and country area are provided for 37 European countries which either are part of, or report data to the EEA, namely Albania (AL), Austria (AT), Bosnia and Herzegovina (BA), Belgium (BE), Bulgaria (BG), Switzerland (CH), Cyprus (CY), Czech Republic (CZ), Germany (DE), Denmark (DK), Estonia (EE), Greece (EL), Spain (ES), Finland (FI), France (FR), Hungary (HR), Ireland (IE), Iceland (IS), Italy (IT), Lithuania (LT), Luxembourg (LU), Latvia (LV), Montenegro (ME), North Macedonia (MK), Malta (MT), Netherlands (NL), Norway (NO), Poland (PL), Romania (RO), Serbia (RS), Sweden (SE), Slovenia (SI), Slovakia (SK), Turkey (TR) and the United Kingdom (UK). Additional results are provided in the Appendix A, including seasonal-scale statistics (Tables A1-A6) and mean monthly profiles (Fig. A1-A2) for rural and urban background stations. Further additional results can be found in the Supplement, including overall statistics for all EEA member countries, figures such as Fig. 2-7 but for urban background stations and a visualization of different methods employed by other studies to reconstruct the $PM_{10}$ concentration field in MERRA-2.

**Table 4.** Overall statistics obtained over the period 2003-2020 across Europe, for CAMSRA (subscript C) and MERRA-2 (subscript M). Statistics are shown both on a daily scale (over all cells and days in the period 2003-2020) and on a monthly scale (weight-averaged by $N$ over all median monthly values). OBS and MOD stand for observational and model concentration, respectively. Reactive gases mixing ratios are expressed in $\mathrm{ppbv}$, aerosol concentrations in $\mathrm{\mu g\,m^{-3}}$ and normalized statistics in %.

| Scale | Pollutant | OBS | $MOD_C$ | $MOD_M$ | $nMB_C$ | $nMB_M$ | $nRMSE_C$ | $nRMSE_M$ | $PCC_C$ | $PCC_M$ | N |
|---|---|---|---|---|---|---|---|---|---|---|---|
| Daily | $O_3$ | 31.0 | 27.2 | 41.7 | -12.4 | 34.2 | 35.7 | 48.4 | 0.61 | 0.53 | 3.04 $10^6$ |
| | $NO_2$ | 5.5 | 6.9 | — | 26.1 | — | 79.2 | — | 0.60 | — | 2.10 $10^6$ |
| | CO | 216.3 | 190.2 | 124.2 | -12.0 | -42.6 | 85.0 | 95.0 | 0.28 | 0.22 | 0.16 $10^6$ |
| | $SO_2$ | 1.6 | 1.7 | 2.2 | 9.5 | 39.5 | 142.6 | 144.6 | 0.33 | 0.35 | 0.77 $10^6$ |
| | $PM_{10}$ | 18.3 | 20.9 | 23.7 | 13.9 | 29.0 | 81.3 | 129.1 | 0.45 | 0.22 | 1.83 $10^6$ |
| | $PM_{2.5}$ | 11.8 | 13.5 | 10.8 | 14.3 | -9.1 | 96.2 | 97.5 | 0.43 | 0.29 | 0.75 $10^6$ |
| Monthly | $O_3$ | 30.3 | 26.6 | 41.7 | -10.0 | 41.9 | 30.0 | 49.5 | 0.53 | 0.23 | 216 |
| | $NO_2$ | 4.7 | 6.9 | — | 41.4 | — | 69.6 | — | 0.48 | — | 216 |
| | CO | 182.0 | 188.2 | 118.9 | 1.8 | -31.2 | 33.9 | 41.6 | 0.53 | 0.55 | 216 |
| | $SO_2$ | 1.3 | 1.3 | 2.2 | -0.2 | 74.5 | 69.7 | 108.4 | 0.28 | 0.31 | 216 |
| | $PM_{10}$ | 17.0 | 20.5 | 20.6 | 18.6 | 32.6 | 59.5 | 86.8 | 0.51 | 0.29 | 216 |
| | $PM_{2.5}$ | 10.3 | 12.9 | 10.4 | 25.1 | 3.7 | 67.7 | 60.5 | 0.51 | 0.48 | 216 |

**Table 5.** Annual trends (Seasonal Theil-Sen estimators, $b$) over the period 2003-2020 across Europe, for rural observations (subscript O), CAMSRA (subscript C) and MERRA-2 (subscript M), together with corresponding 99 % confidence intervals ($\epsilon_-$, $\epsilon_+$). Statistically significant annual trends are highlighted in bold. Trends and uncertainty ranges are expressed in $\mathrm{ppbv\ y^{-1}}$ and $\mathrm{\mu g\,m^{-3}\ y^{-1}}$ for reactive gases and aerosols, respectively. Relative trends (normalized by the mean concentration over 2003-2020) are also indicated in parenthesis.

| Pollutant | $b_O$ | $\epsilon_{O-}$ | $\epsilon_{O+}$ | $b_C$ | $\epsilon_{C-}$ | $\epsilon_{C+}$ | $b_M$ | $\epsilon_{M-}$ | $\epsilon_{M+}$ |
|---|---|---|---|---|---|---|---|---|---|
| $O_3$ | +0.03 (+0.11 %/yr) | -0.26 | +0.22 | **+0.23** (+0.9 %/yr) | +0.01 | +0.46 | -0.06 (-0.15 %/yr) | -0.22 | +0.11 |
| $NO_2$ | **-0.11** (-2.3 %/yr) | -0.17 | -0.07 | **-0.17** (-2.5 %/yr) | -0.23 | -0.12 | — | — | — |
| CO | **-3.47** (-1.9 %/yr) | -5.15 | -2.43 | **-4.56** (-2.4 %/yr) | -6.26 | -3.40 | -0.44 (-0.37 %/yr) | -0.88 | -0.07 |
| $SO_2$ | **-0.034** (-2.7 %/yr) | -0.042 | -0.029 | **-0.078** (-6.2 %/yr) | -0.082 | -0.071 | **-0.033** (-1.5 %/yr) | -0.052 | -0.017 |
| $PM_{10}$ | **-0.36** (-2.1 %/yr) | -0.46 | -0.28 | **-0.70** (-3.3 %/yr) | -0.84 | -0.60 | -0.02 (-0.10 %/yr) | -0.18 | +0.06 |
| $PM_{2.5}$ | -0.10 (-0.91 %/yr) | -0.15 | -0.02 | **-0.23** (-1.7 %/yr) | -0.34 | -0.17 | -0.002 (-0.02 %/yr) | -0.079 | +0.045 |

## 3.1 Ozone ($O_3$)

Overall, CAMSRA reproduces the observed [$O_3$] fairly well, with limited negative bias (-12 %), and reasonable error and correlation (36 % and 0.61, respectively). In comparison, MERRA-2 systematically overestimates [$O_3$] (+34 %) and shows poorer error and correlation (48 % and 0.53, respectively). On average, observed $O_3$ mixing ratios reach a minimum between late autumn and early winter, then peak in spring and are followed by persistently high but slowly decreasing $O_3$ levels until reaching a sharp drop in late summer (Fig. A1 in the Appendix). CAMSRA captures reasonably well the seasonality of $O_3$, although with negative bias during winter and early spring. Conversely, MERRA-2 substantially underestimates the seasonal amplitude (around 15 ppbv, against more than 20 ppbv in observations and CAMSRA).

Throughout the entire period, the median monthly-scale nMB in CAMSRA remains below -20 %, with larger underestimations through the beginning of the period and better results during the last years. The bias displays a clear seasonal pattern, with an important winter and spring deterioration (-21 and -16 %, respectively) but very limited biases in summer and autumn (-4 and -1 %, respectively). Such oscillating biases have also been reported by Huijnen et al. (2020) over Europe. Regarding the other metrics, median monthly-scale nRMSE in CAMSRA reaches its worst values in winter (36 %), when the PCC is conversely the best (0.71), whereas an opposite behaviour with low nRMSE and poor PCC can be observed in summer (26 % and 0.40, respectively). A strong seasonal variability is also found in MERRA-2 statistics, although limited to nMB and nRMSE, which are worst in autumn (+61 % and +67 %,respectively). While the reasonable PCC obtained over the entire dataset (0.53) is likely driven by a good ability of MERRA-2 to capture the $O_3$ seasonality, the much lower monthly PCC values (oscillating around 0.25) suggest that MERRA-2 represents the intra-monthly variability of daily $O_3$ mixing ratios very poorly over a large part of the domain. Nonetheless, MERRA-2 is able to reproduce the spring peak followed by a slow decrease in [$O_3$] typically seen in European observations during summer. In contrast, CAMSRA completely misses this mid-spring $O_3$ peak, as shown in Fig. A1. Over 2003-2020, no statistically significant annual trend (estimated as a Seasonal Theil-Sen slope) of mean [$O_3$] is observed over Europe, neither in MERRA-2 nor in the observations. However, a significant though low positive increase

of +0.23 ppbv y$^{-1}$ is found in CAMSRA (Table 5), at least partly due to the aforementioned stronger underestimation of O$_3$ during the first years of the period.

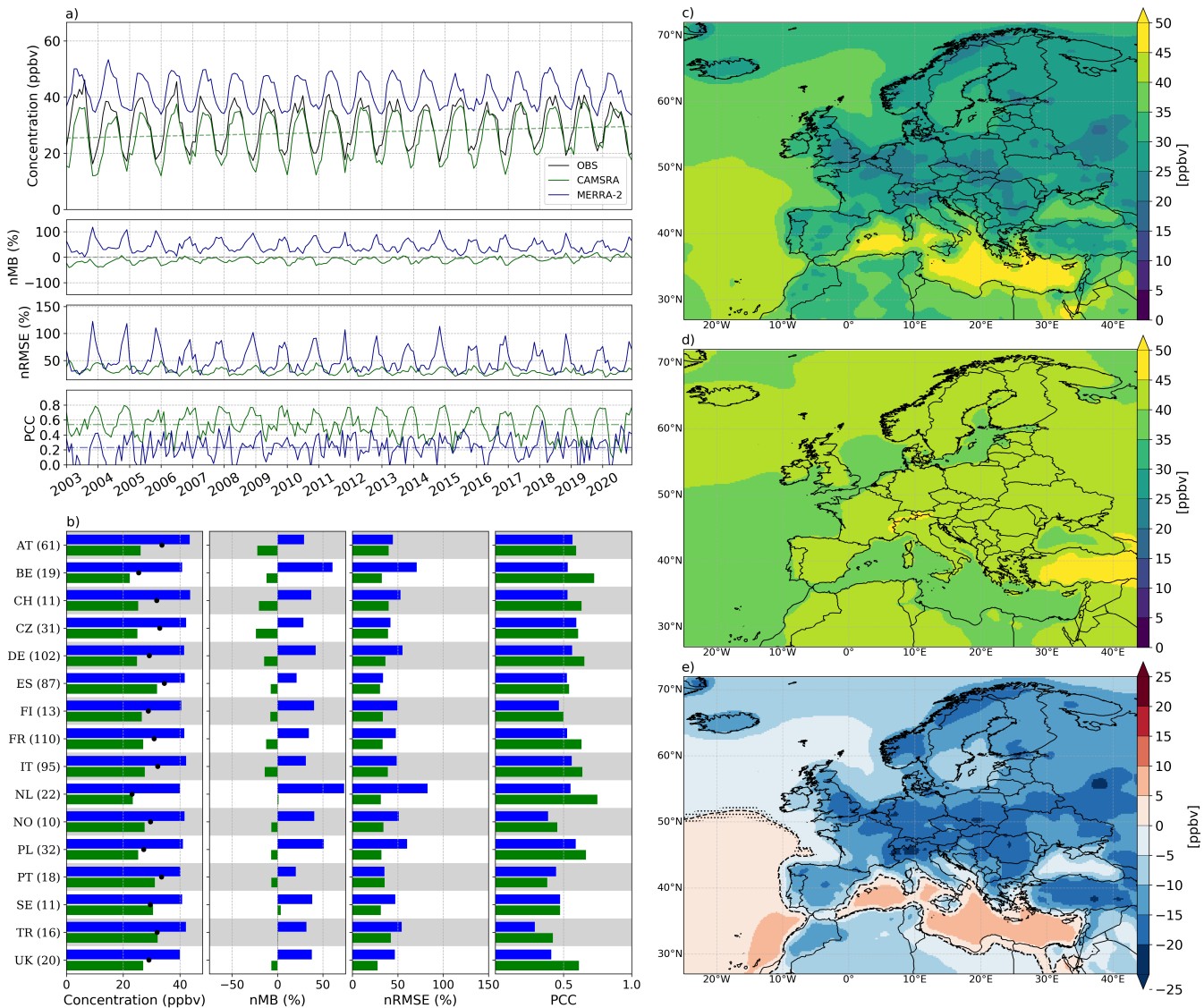

**Figure 2.** Evaluation of $O_3$ over Europe depicting: a) Monthly time series of [$O_3$], nMB, nRMSE and PCC over the period 2003-2020; b) Spatially-averaged [$O_3$], nMB, nRMSE and PCC for countries with at least 5 cells with observations; c) Mean [$O_3$] climatology in CAMSRA; d) Mean [$O_3$] climatology in MERRA-2; e) Differences in Mean [$O_3$] climatology between CAMSRA and MERRA-2. Black, green and blue colors in a) and b) indicate observations, CAMSRA and MERRA-2, respectively. Numbers between parentheses in b) indicate the cells with available observations. Only PCC values in the range 0–1 are displayed in b). Statistically significant trends, at a 99 % confidence level, are displayed in a). Dotted areas in e) indicate where the differences are not statistically significant at a 99 % confidence level, whereas the black dashed contour stands for a zero difference in concentration between reanalyses.

The country-level evaluation highlights how CAMSRA outperforms MERRA-2 in every single country across the European continent for every computed statistic, with the greatest differences appearing in Belgium (BE) and the Netherlands (NL), and the smallest ones in Spain (ES) and Portugal (PT). In CAMSRA the nMB remains generally negative, at around -10 %, with several countries showing virtually no bias (e.g. Netherlands (NL), Turkey (TR), Sweden (SE)), while MERRA-2 displays values in the range +30–70 %. As for the nRMSE, in CAMSRA it remains constrained between 30 and 50 % for all evaluated countries, whereas in MERRA-2 it generally remains close to 50 %, even surpassing this value for several countries, such as the Netherlands (NL), Poland (PL), Belgium (BE) and Turkey (TR). In most countries the PCC does not differ considerably between reanalyses, remaining in the range 0.4–0.7 and slightly higher values for CAMSRA. Despite its greater original resolution, MERRA-2 fails to capture the spatial variability of the $[O_3]$ field, with highly homogeneous mixing ratio values over land, ranging from 35 to 45 ppbv (Fig. 2d), likely a result of the lack of accurate ozone sources in the parameterized chemistry and limited sensitivity of OMI measurements to lower tropospheric ozone (note that neither MLS nor OMI provide ozone profile information in the troposphere). A wider range of assimilated products, as seen in Table 2, and more detailed gas-phase chemistry likely accounts for CAMSRA's better overall performance and greater spatial variability. Nevertheless, we expect the MERRA-2 ozone profile product to be useful for scientific studies that focus on the upper troposphere and the stratosphere, given the high correlations found by Bosilovich et al. (2015) against independent ozonesonde data at these altitudes.

Inness et al. (2019) evaluated surface $O_3$ against the World Meteorological Office (WMO)'s Global Atmosphere Watch (GAW) background stations, and noticed slightly higher negative biases in winter (with modified nMB down to -40%), though based on a different and smaller set of stations (45 GAW stations, against 1511 EEA rural background stations gridded into 728 cells here). Over 2003-2018, Wagner et al. (2021) evaluated CAMSRA surface $O_3$ mixing ratios against European Monitoring and Evaluation Programme (EMEP) observations, finding typically negative modified normalized mean biases (MNMB) within -30 % in winter (driven by underestimated $O_3$ mostly at midlatitudes), but positive ones in summer and autumn, up to +15 %. Such an oscillating bias is in good agreement with our results over the European continent. Although satellite $O_3$ measurements are extensively assimilated in CAMSRA (11 space-based $O_3$ products included), Wagner et al. (2021) already demonstrated their minor impact on surface $O_3$. This may be at least partly due to the relatively low sensitivity of space-borne instruments to lowermost tropospheric $O_3$ (e.g.Cuesta et al. (2013)). All in all, likely due to a more detailed representation of the tropospheric chemistry, CAMSRA clearly outperforms MERRA-2 in simulating surface $O_3$ mixing ratios.

When considering urban background stations (Table B1) the overall nMB in CAMSRA, though shifted in sign, remains very limited (+8 %), whereas MERRA-2 presents an overestimation (+64 %) which nearly doubles the one found in the rural subset. Such an evolution of the statistics at least partly reflects the intrinsic difficulty of coarse reanalyses in representing $O_3$ titration in urban areas. For CAMSRA, the nRMSE shows no significant variation (+34 %), though a slight improvement is found for the PCC (0.72), which represents the best overall correlation across all station subsets and pollutants. Compared to the rural subset, MERRA-2 presents a very similar PCC (0.54), though an important deterioration of the nRMSE is found (+75 %). The overall averaged $[O_3]$ is 5.7 ppbv smaller than in the rural stations subset.

## 3.2 Nitrogen dioxide (NO$_2$)

CAMSRA systematically overestimates the mixing ratio of NO$_2$ (Fig. 3a) throughout the entire period of study, with an overall moderate positive bias of +26 % (Table 4), although the seasonal variability of NO$_2$ is well captured. In contrast, over 2003-2016 Inness et al. (2019) reported mostly limited negative biases, but based on a very small set of regional background stations (4 GAW stations) against 1460 EEA stations gridded into 609 cells in the present study. Overall, CAMSRA shows a relatively large overall nRMSE (79 %) and reasonable PCC (0.60).

At median monthly-scale, biases increase from +12 % in winter to +42 % in summer (Table A2). Monthly-scale nRMSE and PCC values show substantial seasonal variations, with better performance in winter (nRMSE and PCC of 70 % and 0.60, respectively) and a notable deterioration in summer (92 % and 0.45).

In terms of long-term trends, the significant decrease of [NO$_2$] observed over 2003-2020 (-0.11 ppbv y$^{-1}$) is moderately overestimated by the reanalysis (-0.17 ppbv y$^{-1}$, i.e. differing by a 1.5 factor). In relative terms, these decreasing mixing ratio trends found for NO$_2$ in the observations and CAMSRA (-2.3 and -2.5 % y$^{-1}$, respectively) are close to the -2.0 % y$^{-1}$ NOx emission trend reported by the EEA over the period 1990-2019 in its emission inventory report (European Environment Agency, 2021).

Although it has been demonstrated that the COVID-19 pandemic reduced the NO$_2$ levels over Europe in 2020 (Bauwens et al. (2020); Vîrghileanu et al. (2020); Petetin et al. (2020); Barré et al. (2021)), the observed [NO$_2$] time series only shows a limited reduction, given only rural background stations are retained for the evaluation and NO$_2$ is a predominantly urban pollutant. The change in CAMSRA appears less pronounced, potentially due to the coarse resolution of the reanalysis, but most likely due to CAMSRA following the RCP8.5 for emissions after 2010 Granier et al. (2011).

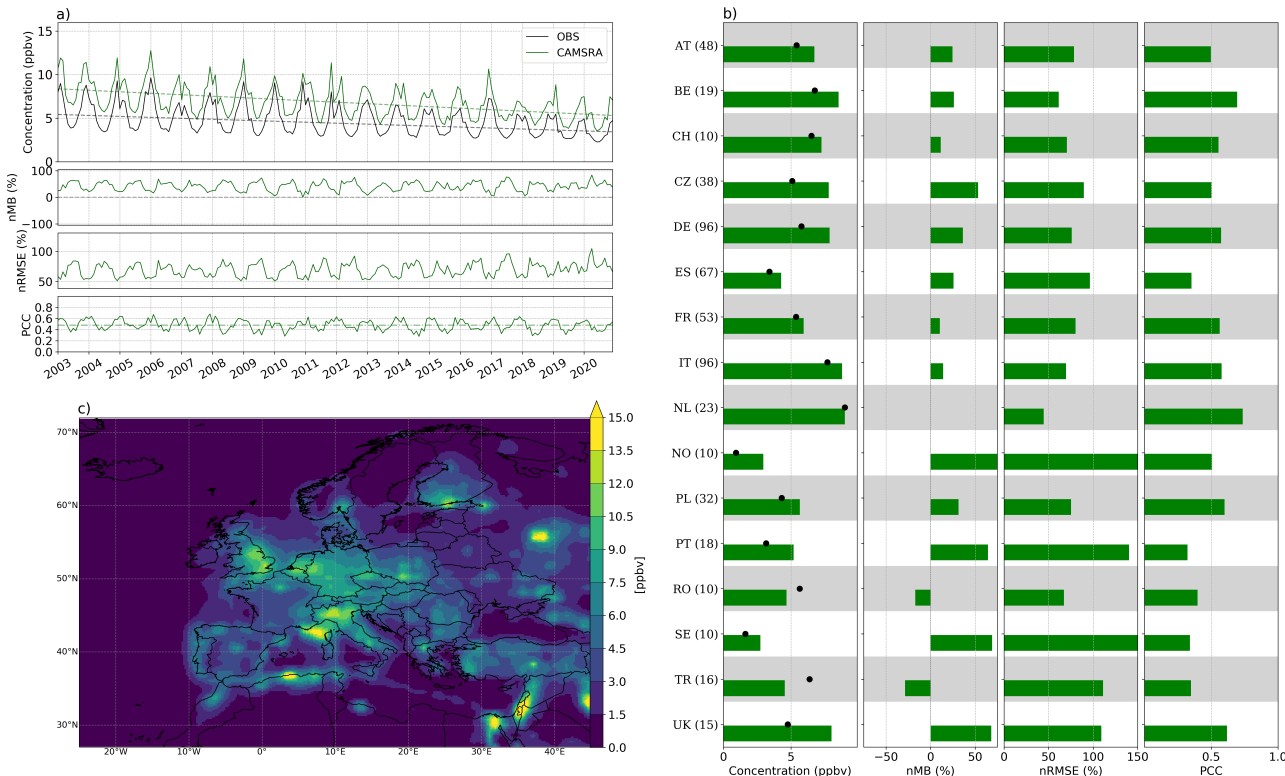

**Figure 3.** Evaluation of NO$_2$ over Europe depicting: a) Monthly time series of [NO$_2$], nMB, nRMSE and PCC over the period 2003-2020; b) Spatially-averaged [NO$_2$], nMB, nRMSE and PCC for countries with at least 5 cells with observations; c) Mean [NO$_2$] climatology in CAMSRA. Black and green colors in a) and b) indicate observations and CAMSRA, respectively. Numbers between parentheses in b) indicate the cells with available observations. Statistically significant trends, at a 99 % confidence level, are displayed in a).

At country-level (considering only countries with more than 5 cells containing observations), most nMBs fall roughly between +10 and +60 %, with the notable exception of Finland (FI) and Turkey (TR), where a moderate underestimation (-15 and -25 %, respectively) is found. The nRMSE ranges from around 60 to over 150 %, depending on the country considered. The

325 PCC remains generally around 0.5, though countries with fewer measuring stations available tend to present lower PCC values (Fig. 3b). Interestingly, virtually no bias is found in the Netherlands (NL), which also displays the lowest error and highest correlation amongst all the countries examined.

The spatial variability of the [NO$_2$] field across the European continent is consistent with the location of dense urban areas (e.g. Paris, Moscow, Barcelona, Oslo, Algiers), highly industrialized regions (e.g. Po River basin, Rhine-Rühr valley, Silesia)

and busy shipping lanes (e.g. Mediterranean, English Channel, Portuguese coastline). In sparsely populated areas, low industrialized regions and the open seas [NO$_2$] levels remain below 3 or even 1.5 ppbv (Fig. 3c).

When considering urban background stations, CAMSRA systematically underestimates [NO$_2$] across the European conti-

nent (Table B1), with an overall strong negative bias (-40 %, Table B1), which can be related in all likelihood to its overly

coarse spatial resolution, that intrinsically prevents a correct representation of urban $NO_2$ hot-spots, as well as to the short

chemical lifetime of $NO_2$. By evaluating $NO_2$ tropospheric columns against satellite-based observations, Inness et al. (2019)

and Wagner et al. (2021) also reported negative biases over Europe, especially during wintertime. Although this contrasts with

the numbers obtained for rural background stations, it is in good agreement with our results for the urban subset, though biases

are significantly larger here (evaluated against 6921 EEA urban background stations, gridded into 1461 cells). The underesti-

mation becomes more critical in winter (-45 %, Table A2) and slightly improves in summer (-33 %). Note that Ryu and Min

(2021) also found a large underestimation of $NO_2$ in winter over South Korea (around -10 ppbv, against -2 ppbv in summer).

CAMSRA also displays a large nRMSE and moderate PCC (68 % and 0.56, respectively). The seasonality and intra-annual

variability of the $NO_2$ mixing ratio fields are both well captured by CAMSRA.

### 3.3  Carbon monoxide (CO)

As shown in Fig. 4a, MERRA-2 systematically underestimates the mixing ratio of CO (overall nMB of -43 %), while CAMSRA

reproduces the observed mixing ratio well, with overall limited mean bias (-12 %). MERRA-2 dramatically fails at reproduc-

ing the seasonal variability of CO, with strongest negative biases in winter (-51 %). Conversely, CAMSRA captures well the

seasonal cycle, although negative biases are also somewhat stronger in winter (-15 %). Note that Ryu and Min (2021), in its

evaluation over South Korea, also reported a severe winter underestimation in CAMSRA together with an absence of variability

in surface CO over the period 2003-2018 in MERRA-2. Interestingly, CAMSRA displays a lack of nMB seasonality, with an

almost constant value throughout summer, autumn and winter. A likely explanation for this is the good ability of CAMSRA

to capture the intra-annual variability of [CO] throughout the year. The overall nRMSE is high in both reanalyses (85 and 95

%, respectively), with again a lower winter performance in MERRA-2 and an overall absence of seasonality in CAMSRA.

Wagner et al. (2021) evaluated CO in Europe against data from GAW stations over the period 2003-2018, reporting a persistent

underestimation (modified nMB ranging from -10 to -20 %) of surface CO, in agreement with our results. In contrast, Inness

et al. (2019) reported an overall overestimation of around 10 ppbv for the period 2003-2017, which again could be due to

the different set of stations taken into account (15 GAW stations, most of them regional and several of them located at high

altitude).

At monthly-scale, the median [CO], nMB and nRMSE in CAMSRA partially capture the seasonality, showing a better per-

formance in autumn (0 %) and summer (31 %), and a moderate springtime (+9 %) and wintertime (39 %) deterioration,

respectively. As seen for $O_3$, the PCC follows the opposite behaviour, with better performance in DJF (0.58) and a late spring-

time deterioration (0.46). In contrast to CAMSRA, MERRA-2 is unable to reproduce the seasonal variability of surface [CO],

despite the nMB and nRMSE displaying significant variability throughout the different seasons. A surprisingly large increase

of [CO] is found in MERRA-2 throughout 2020. It is unclear what stands behind such a significant increase, but this abrupt

change affects mostly specific pollution hotspots in the European continent, including the Rhine-Rühr valley, the Paris and

London metropolitan areas, as well as the Po River basin. This [CO] surge is also found in the raw version (i.e. non-regridded)

of the reanalysis. The strong statistically significant decrease in CO observed across Europe over 2003-2020 (-3.47 ppbv $y^{-1}$)

is moderately overestimated in CAMSRA (-4.56 ppbv y$^{-1}$), although less dramatically than in MERRA-2, where CO remains roughly constant over all the period study, displaying a small negative trend (-0.44 ppbv y$^{-1}$). In European Environment

Agency (2021) the EEA reports a CO emission trend of -2.3 % y$^{-1}$ over 1990-2019, relatively close to the mixing ratio trends found in CAMSRA, -2.4 % y$^{-1}$, and the observations, -1.9 % y$^{-1}$. In 2020, MERRA-2 shows a very large increase of [CO] across most of Europe, in contrast to both CAMSRA and the observations. The overall PCC in MERRA-2 and CAMSRA is poor (0.22 and 0.28, respectively), although better PCC values (~0.40) are found at monthly-scale (0.53 and 0.55, respectively).

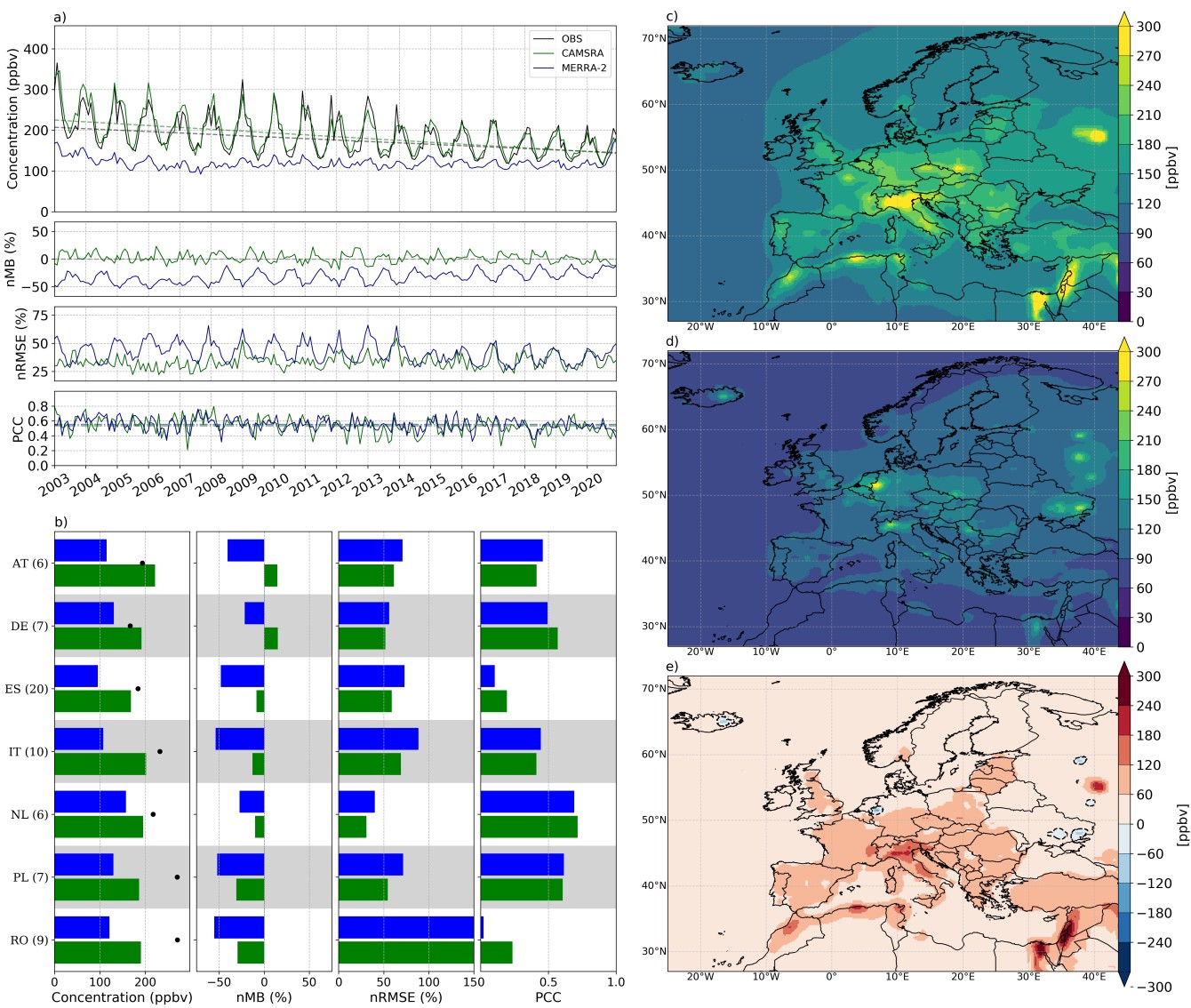

**Figure 4.** Similar to Fig. 2 but for CO.

This CO underestimation typically spreads over all the European continent, with strong differences across countries. As CO is not assimilated in MERRA-2, but simulated by the GEOS-5 modeling system, this underestimation likely comes from a poor representation of CO emissions and/or excessively large CO sinks. In both reanalyses, the best scores in terms of bias, PCC and nRMSE are found in Germany (DE), and to a lesser extent in the Netherlands (NL). Conversely, far poorer results are obtained in Poland (PL) and Romania (RO). Although different, the nMB and nRMSE in both reanalyses typically show comparable variations from one country to another. Both CAMSRA and MERRA-2 show CO hot-spots over large urban areas and/or highly industrialised regions (e.g. Moscow, Po River basin). However, compared to CAMSRA, MERRA-2 highlights some additional hot-spots, for instance on the Vatnajökull ice cap, located in Iceland, a region well known for its sub-glacial volcanoes (e.g. Grímsvötn) which experience frequent degassing. Another significant hot-spot is found in the Donets Basin (eastern Ukraine), an important coal-mining region. Two other CO hot-spots can be seen south and north of Moscow, corresponding to the cities of Voronezh and Yaroslavl, respectively, but it is unlikely that CO levels comparable to those of Moscow are found in these intermediate sized cities (Fig. 4c,d).

The reanalyses also differ in the locations where [CO] is higher across Europe (Po River basin in CAMSRA; Rhine-Rühr valley in MERRA-2). CAMSRA highlights the highest CO mixing ratios in Europe in the Po River basin and displays moderate mixing ratio values in the Rhine-Rühr area, which suggests a longer CO lifetime in the former given that European Environment Agency (2021) reports the highest CO emissions, over all the period 1990-2019, in Germany. Therefore, in sharp contrast with CAMSRA, MERRA-2 obviously fails to capture the chemistry processes of surface CO, with a likely underestimation of emission sources and/or too large CO sinks, thus being unable to reproduce the spatiotemporal variability of surface CO observed over Europe.

From Table B1 it immediately becomes apparent that the main difference between the urban and rural subsets, aside from the large variation in baseline mixing ratios, comes from CAMSRA largely underestimating the observed [CO] in urban cells, with the nMB (-46 %) nearly quadrupling when compared to the rural evaluation. For MERRA-2 the nMB also suffers a deterioration (-64 %), but more limited due to an already large bias in the rural subset. For both CAMSRA and MERRA-2, the overall nRMSE (91 and 105 %, respectively) and PCC (0.39 and 0.19, respectively) remain close to the rural values, with no significant variations. The seasonal behaviour of both reanalyses also remains unchanged, with MERRA-2 completely missing the amplitude of the seasonal cycle. This large amplitude is also the reason why CAMSRA loses its ability to reproduce the observed CO mixing ratio.

### 3.4 Sulphur dioxide (SO$_2$)

When computed over the entire dataset (Table 4), the statistics of CAMSRA and MERRA-2 show very poor nRMSE and PCC (around 143 % and 0.33–0.35, respectively), but better performance in terms of bias for CAMSRA (+10 %) than for MERRA-2 (+40 %). On average, the overestimation of MERRA-2 is much higher in winter, meaning the amplitude of the SO$_2$ seasonal cycle is strongly overestimated (Fig. A1).

At monthly-scale (Fig. 5a), the median nMB in MERRA-2 severely deteriorates (+75 %) and increases throughout time, with the worst performance peaking in SON (+94 %) and a slight springtime improvement (+57 %). The median monthly-scale nMB in CAMSRA tends to improve between late spring and early summer, reaching values close to 0 %, though it oscillates throughout the year, dropping to -12 % in winter and peaking at +11 % in autumn. Note that Ryu and Min (2021), though finding a larger [$SO_2$] overestimation over South Korea, greater than the underestimation shown here for Europe, found a similar nMB seasonality, with nMB improving (~+2 ppbv) and worsening (~+6 ppbv) in warm and cold months, respectively. In MERRA-2 the median nMB oscillates roughly around +69 % (with a $\pm$ 3 % range), though it suffers an important increase (with significant intra-annual variability) from 2013 onwards due to a decrease in observed [$SO_2$]. A similar increase is also observed for the nRMSE. The monthly-scale nRMSE and PCC remain roughly constant (when averaged across all months) throughout all seasons, both in CAMSRA (around 70 % and 0.28, respectively) and in MERRA-2 (around 108 % and 0.31, respectively), though the latter displays much stronger seasonal variability. Note also the large difference between the monthly-scale nRMSE (70–108 %) and the overall nRMSE (around 143 %). The statistically significant negative trend found in observed $SO_2$ mixing ratios (-0.034 ppbv y$^{-1}$) is largely overestimated by CAMSRA (-0.078 ppbv y$^{-1}$), and well reproduced by MERRA-2 (-0.033 ppbv y$^{-1}$) (Fig. 5). In European Environment Agency (2021) the EEA reports a $SO_2$ anthropogenic emission trend of -3.2 % y$^{-1}$ over 1990-2019, falling between the mixing ratio trend found in CAMSRA, -6.2 % y$^{-1}$, and the one found in the observations, -2.7 % y$^{-1}$, and MERRA-2, -1.5 % y$^{-1}$.

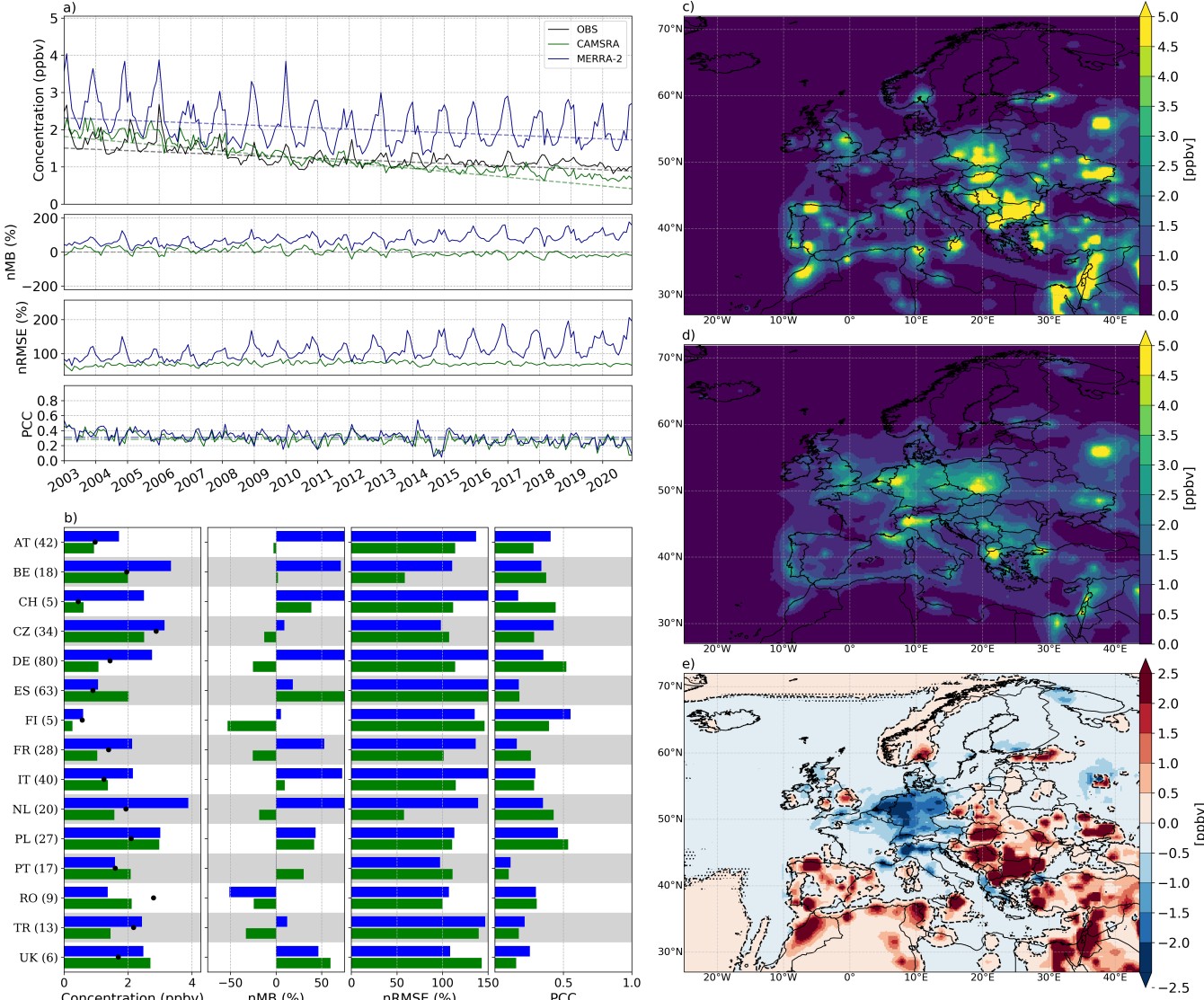

**Figure 5.** Similar to Fig. 2 but for SO$_2$.

The country-level evaluation for SO$_2$ shows very heterogeneous results across countries, differing substantially from the observed behaviour in previously examined reactive gases. The nMB presents a wide range of variation, with certain countries 425 showing very reduced biases for at least one of the reanalysis (e.g. Portugal, Czechia, Austria, Belgium) and others presenting biases well over ± 50 % (e.g. United Kingdom, France, Romania, Switzerland). Both the nRMSE and PCC display a poor performance, ranging roughly within 100–150 % and 0.10–0.50, respectively (Fig. 5b). On a first examination of the SO$_2$ spatial distribution, it may appear as if the mixing ratio values in the time series should be larger for CAMSRA, though this is actually

misleading as the evaluation is performed only in cells with available observations. Therefore, regions with a higher station density contribute more towards the final mixing ratio value. From Fig. 5e it can be immediately seen that MERRA-2 presents higher $SO_2$ mixing ratios in several countries which have an overall larger number of stations (e.g. Germany, Netherlands, France, Italy).

In both reanalyses, the heterogeneous distribution of $[SO_2]$ is consistent with the location of highly industrialized areas (e.g. Po River basin, Rhine-Rühr valley) and coal-mining regions (e.g. Silesia, Donets Basin, Balkans). To a minor extent, there are also significant $SO_2$ mixing ratios in dense urban areas and along shipping lanes. Surprisingly, the aforementioned CO hot-spot found in MERRA-2 over the Icelandic Vatnajökull ice cap does not come with an associated $SO_2$ hot-spot, which contrasts with the fact that $SO_2$ emissions represent a large fraction of volcanic gases. The reanalyses show sharp differences in the regions where highest mixing ratios of $SO_2$ are present, with CAMSRA favouring coal-mining regions and dense urban areas, and MERRA-2 showing a more balanced distribution between them (Fig 5c,d,e). Overall, both reanalysis products present distinct although substantial deficiencies in their representation of $SO_2$ mixing ratios, with the increasing overestimation of MERRA-2 being probably the most critical issue. Anthropogenic $SO_2$ emissions in MERRA-2 are obtained from AeroCom Phase II (Diehl et al. (2012)) and EDGAR v4.2 (European Commission, 2011 [https://edgar.jrc.ec.europa.eu/]) inventories, with emissions fixed to those of the last year available in each inventory (Randles et al., 2017). Thus, the progressive deterioration of the bias in MERRA-2, particularly notorious from 2013 onwards, likely arises due to an emission overestimation which propagates throughout the time period where no updated $SO_2$ emissions are available.

When considering urban background stations, both CAMSRA and MERRA-2 shift towards a moderate negative nMB (-29 and -26 %, respectively), far from the positive bias found in the rural subset. Overall, both the nRMSE (247 and 251 %, respectively) and PCC (0.18 and 0.08, respectively) are extremely poor (see Table B1). The mixing ratio in CAMSRA presents significant intra-annual variability and thus fails to correctly reproduce the observed seasonal behaviour. MERRA-2 shows a much better ability to capture the seasonality of $[SO_2]$, though it still suffers from the increasing overestimation previously highlighted for rural background stations.

### 3.5 Coarse particulate matter ($PM_{10}$)

Overall, CAMSRA and MERRA-2 reanalyses represent moderately well surface $PM_{10}$ concentrations over Europe (Table 4), with a limited positive nMB (+14 %) for CAMSRA and moderate bias for MERRA-2 (+29 %), but poor nRMSE (81 and 129 %, respectively) and PCC (0.45 and 0.22, respectively).

At monthly-scale, the median nMB in CAMSRA presents a strong seasonality, with an important deterioration during spring (+36 %) and better performance in DJF (+5 %), while the nRMSE and PCC show a strong and complex intra-annual variability without a clear seasonal pattern (remaining in the range 53–65 % and 0.48–0.54, respectively). In comparison, nRMSE and PCC in MERRA-2 follow a clear seasonal behaviour, with strongly deteriorated results during winter (105 % and 0.11, respectively) but better summertime performance (71 % and 0.41, respectively). Surprisingly, the median nMB in MERRA-2 also peaks in JJA (+38 %), with a small bias reduction in SON, and a wintertime low (+24 %). Ryu and Min (2021) found a

slightly positive $PM_{10}$ bias for CAMSRA in South Korea over 2003-2018, while for MERRA-2 their findings suggest a clear underestimation that worsens significantly in winter, the former being in good agreement with our results over Europe. The statistically significant negative trend present in the observations (-0.36 $\mu g\,m^{-3}\,y^{-1}$) is strongly overestimated by CAMSRA (-0.70 $\mu g\,m^{-3}\,y^{-1}$) and severely underestimated by MERRA-2 (-0.02 $\mu g\,m^{-3}\,y^{-1}$), with the latter not being statistically significant (at a 99 % confidence level). In European Environment Agency (2021) the EEA reports a $PM_{10}$ emission trend of -1.7 % $y^{-1}$ over 2000-2019, far from the concentration trend of CAMSRA, -3.3 % $y^{-1}$, but closer to the one found in the observations, -2.1 % $y^{-1}$.

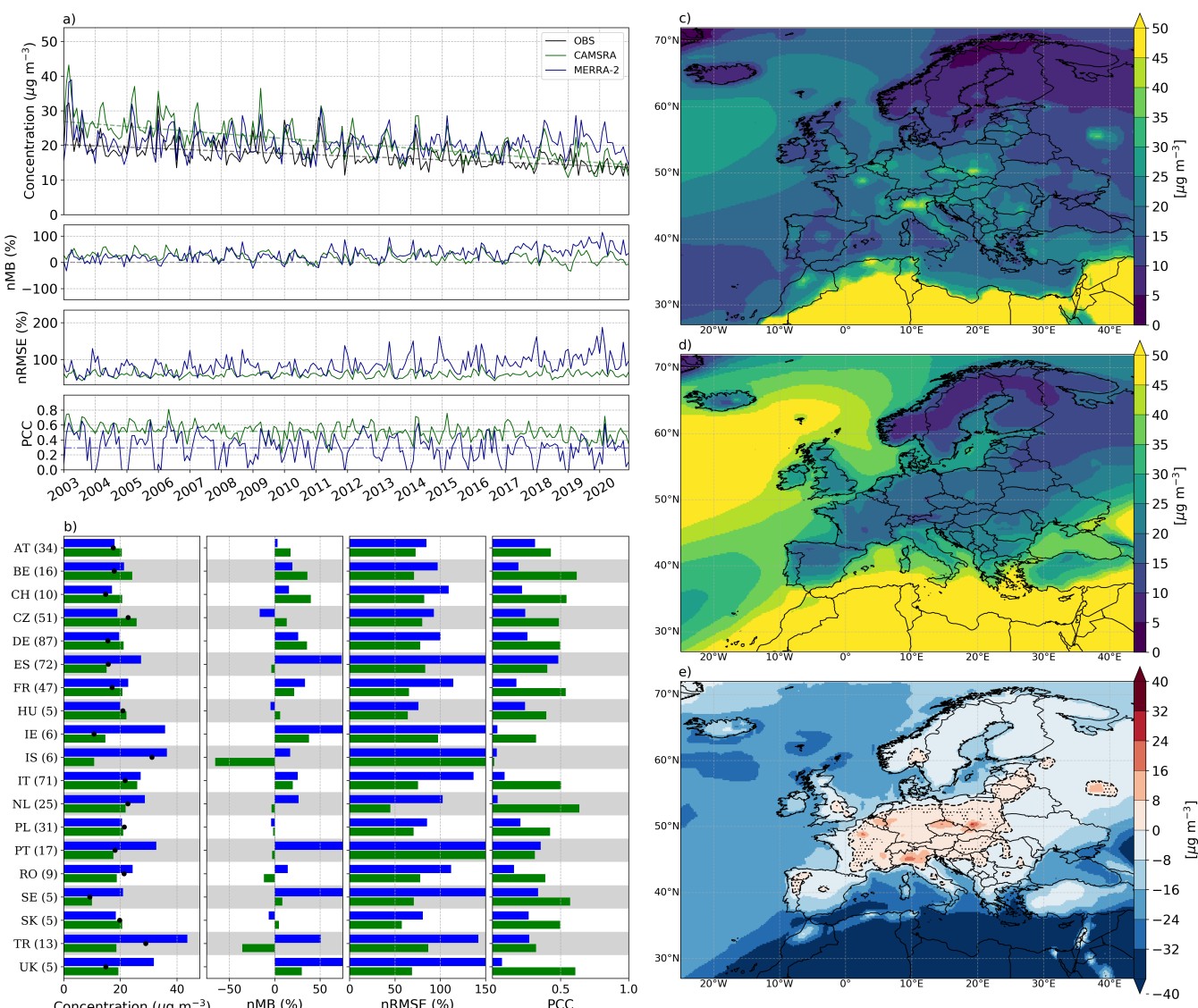

**Figure 6.** Similar to Fig. 2 but for $PM_{10}$.

At country-level, CAMSRA tends to outperform MERRA-2 in most countries, with lower nRMSE (50–100 % and 75–150 %, respectively) and higher PCC values (0.3–0.6 against 0.1–0.4, respectively). The nMB presents a wide range of variation in both reanalyses, with certain countries showing virtually no bias for MERRA-2 (e.g. Austria), for CAMSRA (e.g. Spain, Netherlands, Portugal) or for both reanalyses (e.g. Poland, Hungary, Slovakia). Other countries present biases well over $\pm$ 25 % (e.g. Turkey, Germany, Ireland, United Kingdom). Though MERRA-2 presents lower nMB values than CAMSRA in several countries (e.g. Iceland, Germany, Czechia, Belgium), both the nRMSE and PCC point towards a greater performance by CAMSRA in all cases (Fig. 6b).

Again, despite its finer resolution, MERRA-2 displays a more homogeneous concentration over land in which the multiple $PM_{10}$ hotspots found in CAMSRA - in industrialized regions (e.g. Po River basin, Silesia) and in certain urban areas (e.g. Paris, Moscow, Madrid) - are missing. In addition, it also shows much higher $PM_{10}$ concentrations over the open seas and Northern Africa, where sea salt and dust sources are predominant. It thus seems that Eq. 2a severely overestimates the surface concentrations of $PM_{10}$, as shown in Fig. 6d), with MERRA-2 displaying differences of more than a 100 $\mu g\,m^{-3}$, particularly over desert areas. This overestimation is likely related to sea salt and dust concentrations in the model being overestimated, as it is shown in the Supplement. Overall, CAMSRA unambiguously outperforms MERRA-2 in capturing the spatiotemporal variability of $PM_{10}$ surface concentrations over Europe.

As shown in Table B1, both CAMSRA and MERRA-2 present limited negative nMB (-20 and -8 %, respectively) for the urban subset, which contrasts with the positive bias found for rural stations. For both reanalyses, the overall nRMSE (85 and 112 %, respectively) and PCC (0.36 and 0.19, respectively) remain close to their rural counterparts, with no significant variations. The observed $PM_{10}$ concentration is characterized by strong intra-annual variability, though certain seasonality is still present.

## 3.6 Fine particulate matter ($PM_{2.5}$)

MERRA-2 reproduces moderately well surface $PM_{2.5}$ concentrations over Europe (Table 4), with a low negative nMB (-9 %) but poor nRMSE and PCC (98 % and 0.29, respectively), while CAMSRA presents an overall worst nMB (+14 %), similar nRMSE (96 %) and slightly better but still moderate PCC (0.43).

The median monthly-scale nMB in CAMSRA presents a clear seasonal pattern, with the bias heavily deteriorating in MAM and JJA (+41 %) but virtually vanishing in DJF (+1 %). MERRA-2 also shows a clear seasonality, with the largest over- and underestimations ocurring during summer (+21 %) and winter (-17 %), respectively. Interestingly, the MERRA-2 and CAMSRA nMB time series, while initially displaying an absolute difference of ~50 %, converge from 2017 onwards. Similarly to the behaviour observed for $PM_{10}$, the median nRMSE and PCC in CAMSRA show a strong intra-annual variability without a clear seasonal pattern (remaining in the range 61–74 % and 0.48–0.53, respectively). As for MERRA-2, both the nRMSE and the PCC present significant seasonal variability, with better performance in summer (50 % and 0.58, respectively) and a heavy wintertime deterioration (74 % and 0.36, respectively). Similar results are reported by Provençal et al. (2017a) when

evaluating MERRA-1 over Europe, with an overall limited negative bias and a deterioration in winter. Note also that Navinya et al. (2020) evaluated $PM_{2.5}$ in MERRA-2 against 20 background stations in India, finding a moderate negative nMB (-34 % ; -27 $\mu g\,m^{-3}$) and a larger wintertime underestimation, in agreement with our results over Europe. The negative trend present in the observations (-0.10 $\mu g\,m^{-3}\ y^{-1}$) has been found to not be statistically significant, though it is strongly overestimated by CAMSRA (-0.23 $\mu g\,m^{-3}\ y^{-1}$), and completely missed by MERRA-2. As a consequence, though the nMB time series of CAMSRA and MERRA-2 differ by more than 30 % in 2003, they end up converging progressively along the period 2003-2020. In European Environment Agency (2021) the EEA reports a $PM_{2.5}$ emission trend of -1.9 % $y^{-1}$ over 2000-2019 which, while not directly comparable to a concentration trend as previously mentioned, is close to the trend found in CAMSRA, -1.7 % $y^{-1}$, but far from the one found in the observations, -0.9 % $y^{-1}$.

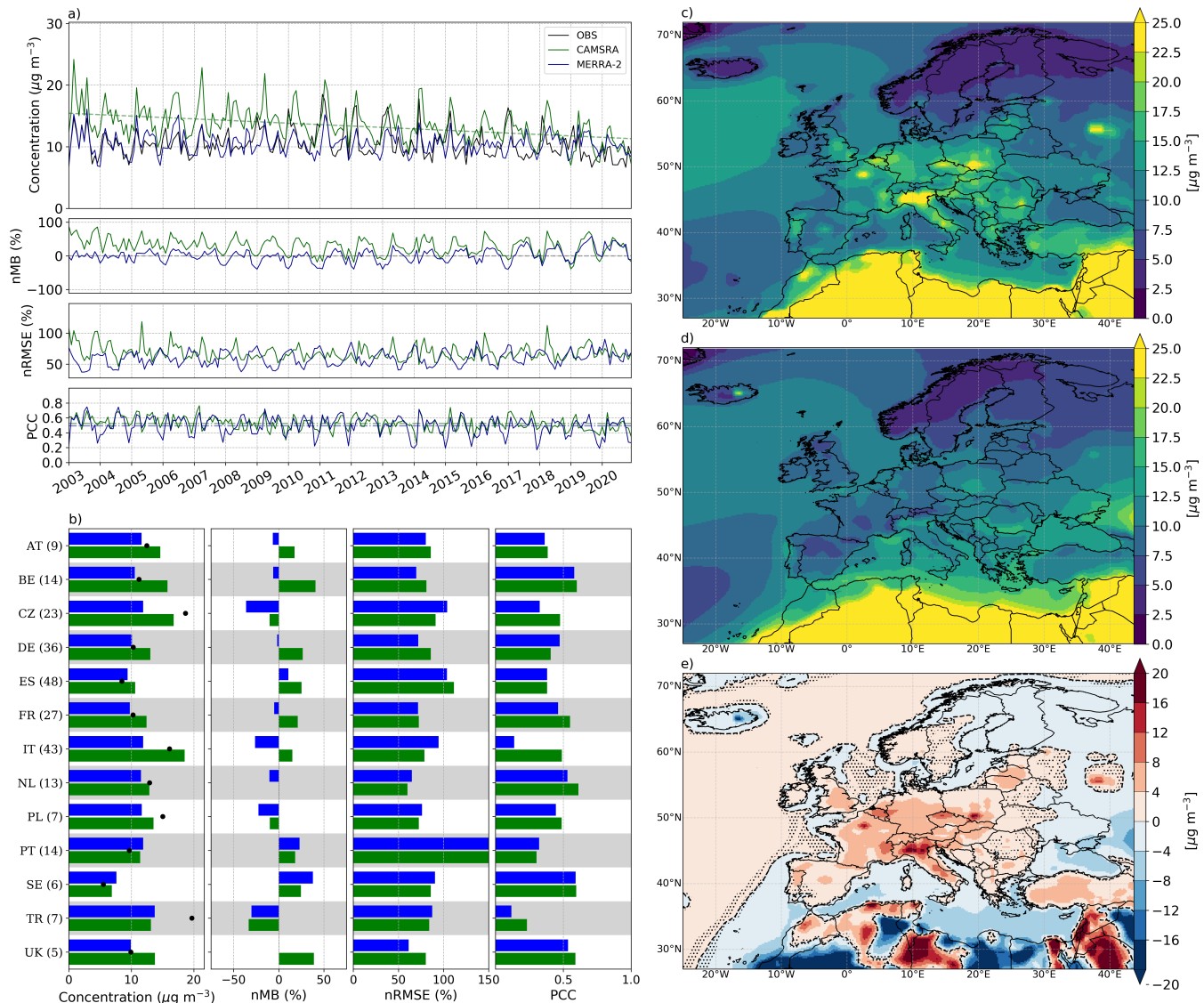

**Figure 7.** Similar to Fig. 2 but for PM$_{2.5}$.

At country-level (Fig. 7b), the differences in PM$_{2.5}$ between CAMSRA and MERRA-2 are less pronounced than for PM$_{10}$, especially for the PCC (with most values in the range 0.3–0.6), and to a lesser extent for the nRMSE (with most values in the range 60–100 %). The nMB presents a similar behaviour to the one observed for PM$_{10}$, with certain countries showing virtually no bias for CAMSRA (e.g. Netherlands) or MERRA-2 (e.g. United Kingdom, France, Germany, Belgium) and other countries presenting important negative/positive biases (e.g. Turkey, Sweden).

The spatial variability of PM$_{2.5}$ concentration remains close to the one obtained for PM$_{10}$ in all regions and in both reanalyses,

except over the open seas, where MERRA-2 no longer shows exceedingly large sea salt levels (which thus prevail mostly in the coarse mode). The surface pollution hot-spots present in Fig. 7 are essentially the same ones that appear in Fig. 6, though a notable exception is observed in MERRA-2 over Iceland. A large $PM_{2.5}$ concentration peak, also visible for $PM_{10}$, can be spotted in Iceland's time series during 2010, surpassing $100\,\mu g\,m^{-3}$, likely due to the Eyjafjallajökull volcanic eruption, which emitted very large amounts of volcanic ash (Thorsteinsson et al., 2012).

As for urban background stations, CAMSRA presents an overall small negative nMB (-13 %) while MERRA-2 displays a larger but limited negative bias (-30 %). In terms of nRMSE and PCC, both CAMSRA and MERRA-2 perform rather poorly, with large errors (86 and 96 %, respectively) and low correlations (0.41 and 0.24, respectively). Similarly to $PM_{10}$, the observed $PM_{2.5}$ concentration shows strong intra-annual variability, though a seasonal pattern is also visible.

## 4 Summary and conclusions

In this work we have performed a long-term (2003-2020) multi-pollutant evaluation of CAMSRA and MERRA-2 global atmospheric composition reanalyses against in situ surface measurements over the European continent. In contrast to past evaluation studies, we have included a more extended set of rural background stations, from several hundred to a few thousand depending on the pollutant considered (Table 3), quality-assured using GHOST metadata and gridded in order to limit, to some extent, representativeness issues. Results obtained against urban background stations have also been briefly discussed.

As a summary, CAMSRA unambiguously outperforms MERRA-2 in representing surface pollutant concentrations across Europe. Differences are particularly clear for $O_3$ and CO, but also persist for $PM_{10}$ and $PM_{2.5}$. CAMSRA clearly achieves the best results for $O_3$, while statistics for the other pollutants show more mixed results: substantial overestimation, moderate error but reasonable correlation for $NO_2$, low biases, poor error and moderate correlation for $PM_{10}$ and $PM_{2.5}$, low biases but poor errors and correlations for CO and $SO_2$. With MERRA-2 being designed mainly for research on aerosols, the reanalysis indeed provides statistics on $PM_{10}$ and $PM_{2.5}$ in line with CAMSRA, but the latter still gives slightly better results over Europe, especially for $PM_{10}$, with overall lower biases and a better characterization of its spatial variability.

Compared to CAMSRA, MERRA-2 benefits from a slightly finer spatial resolution, but assimilates a much less diversified set of satellite products. However, recent evaluations of CAMSRA have noticed that this assimilation only partially improves the representation of pollutant concentrations at the surface, despite a clear improvement being found in the entire troposphere. Although at least partly due to the still coarse spatial resolution of CAMSRA, a large if not dominant part of the model-versus-observation differences found here at the surface are likely explained by errors in emissions and/or sinks. Therefore these global reanalysis datasets need to be carefully bias-corrected with surface observations in order to be used in long-term air pollution and impact studies.

The surface pollution evaluation carried out in this work can serve as a milestone for future air quality and other pollution-related studies. In that regard, further advancements in the field could focus on developing new statistical approaches to merge

surface observations with reanalysis data. As global atmospheric composition reanalyses do not assimilate data at the surface, ground level measurements can be employed, through different statistical methods, to bias correct and improve raw model output statistics, thus leading to more robust reanalysis products. This improved characterization of the spatiotemporal variability of surface air pollution would open the door to improved health impact and air quality assessments, while also helping design and implement more effective air pollution reduction policies.

Eventually, if reanalyses are to be used in long-term health impact studies, consistent statistical approaches to combine observational data with reanalysis data need to be further developed.

*Data availability.* The observational data, obtained from EEA AIRBASE and AQ e-Reporting air quality datasets, and reanalysis data, obtained from CAMSRA and MERRA-2, used in this study are publicly available. CAMSRA, MERRA-2 and EEA observational data can be obtained respectively from the Atmosphere Data Store (ADS; https://atmosphere.copernicus.eu/data), the NASA Goddard Earth Sciences Data and Information Services Center (GES DISC; https://disc.gsfc.nasa.gov/datasets?project=MERRA-2) and from the European Environment Agency websites for AQ e-Reporting (https://www.eea.europa.eu/data-and-maps/data/aqereporting-9) and AIRBASE (https://www.eea.europa.eu/data-and-maps/data/airbase-the-european-air-quality-database-8).

**Appendix A: Seasonal cycle**

Seasonal-scale statistics (Tables A1-A6) and mean monthly profiles (Fig. A1-A2) are shown here for rural and urban background stations.

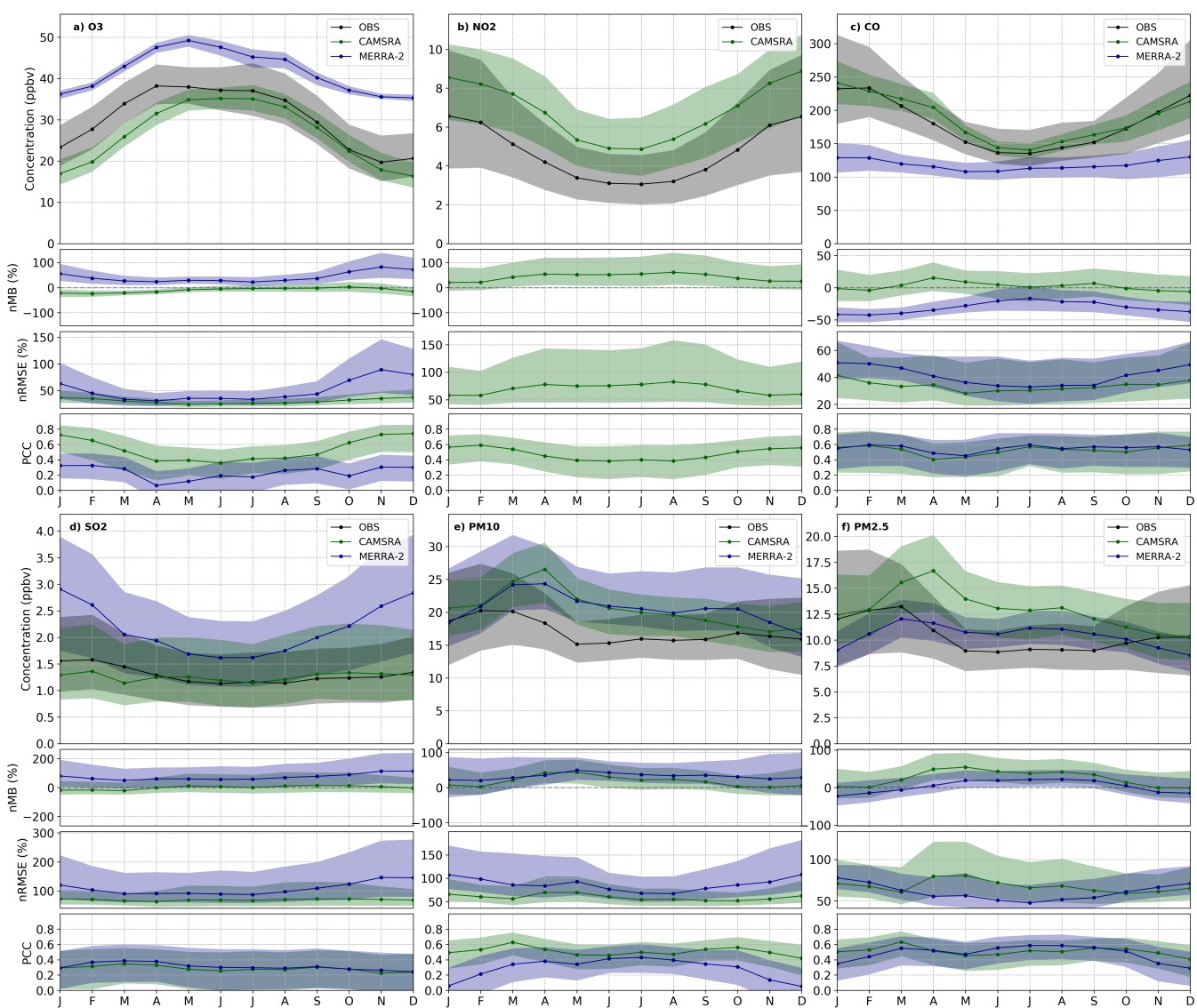

**Figure A1.** Seasonal variability of [O₃], [NO₂], [CO], [SO₂], [PM₁₀] and [PM₂.₅] over the period 2003-2020 across Europe evaluated against rural background stations. For each pollutant the panels show, from top to bottom, concentration, nMB, nRMSE and PCC. The black, green and blue lines represent observations, CAMSRA and MERRA-2, respectively. Shaded contours indicate the 25th (bottom) and 75th (top) percentiles. All monthly values are weighted by the number of points, N, over the period 2003-2020.

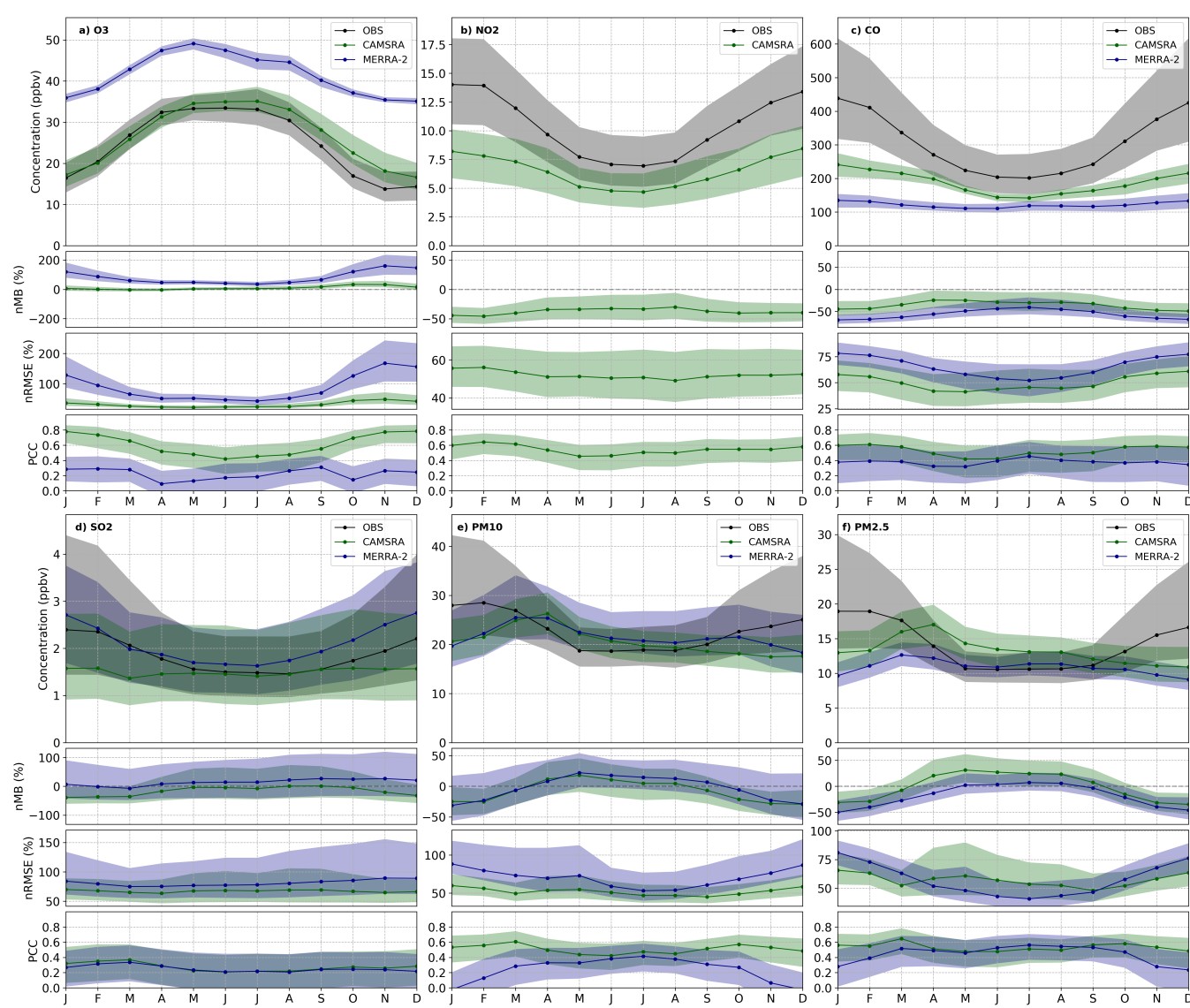

**Figure A2.** Same as Fig. A1 but for urban background stations.

**Table A1.** $O_3$ seasonal statistics over the period 2003-2020 across Europe, for CAMSRA (subscript C) and MERRA-2 (subscript M). Statistics are shown both on a daily scale (d; over all cells and days in the period 2003-2020) and on a monthly scale (m; weight-averaged over all median monthly values). Reactive gases concentrations are expressed in ppbv and normalized statistics in %.

| Type | Scale | Season | OBS | $MOD_C$ | $MOD_M$ | $nMB_C$ | $nMB_M$ | $nRMSE_C$ | $nRMSE_M$ | $PCC_C$ | $PCC_M$ | N |
|------|-------|--------|-----|---------|---------|---------|---------|-----------|-----------|---------|---------|---|
| RUR | Daily | MAM | 37.1 | 31.0 | 46.5 | -16.4 | 25.2 | 32.4 | 37.0 | 0.38 | 0.24 | $0.77\ 10^6$ |
| | | JJA | 37.0 | 34.9 | 45.6 | -5.5 | 23.3 | 30.5 | 38.1 | 0.40 | 0.33 | $0.78\ 10^6$ |
| | | SON | 25.2 | 23.9 | 37.8 | -4.9 | 49.9 | 38.0 | 64.5 | 0.57 | 0.38 | $0.76\ 10^6$ |
| | | DJF | 24.5 | 18.3 | 36.5 | -25.1 | 49.2 | 46.1 | 65.0 | 0.55 | 0.26 | $0.74\ 10^6$ |
| RUR | Monthly | MAM | 36.7 | 30.8 | 46.6 | -15.5 | 26.4 | 27.0 | 33.2 | 0.43 | 0.15 | 54 |
| | | JJA | 36.3 | 34.5 | 45.8 | -3.9 | 26.4 | 25.5 | 35.6 | 0.40 | 0.21 | 54 |
| | | SON | 24.0 | 22.8 | 37.7 | -0.8 | 60.5 | 31.8 | 67.3 | 0.61 | 0.26 | 54 |
| | | DJF | 23.9 | 17.7 | 36.6 | -20.5 | 55.0 | 36.1 | 62.6 | 0.71 | 0.32 | 54 |
| URB | Daily | MAM | 31.1 | 30.9 | 46.4 | -0.6 | 49.3 | 28.0 | 57.8 | 0.52 | 0.24 | $1.30\ 10^6$ |
| | | JJA | 32.9 | 35.1 | 45.6 | 6.8 | 38.7 | 29.5 | 49.4 | 0.46 | 0.22 | $1.31\ 10^6$ |
| | | SON | 19.4 | 24.3 | 37.6 | 25.2 | 93.9 | 45.3 | 105.6 | 0.71 | 0.31 | $1.28\ 10^6$ |
| | | DJF | 17.5 | 18.6 | 36.3 | 6.6 | 107.5 | 42.1 | 121.0 | 0.70 | 0.21 | $1.25\ 10^6$ |
| URB | Monthly | MAM | 30.9 | 30.6 | 46.5 | -1.0 | 51.4 | 23.9 | 56.5 | 0.55 | 0.17 | 54 |
| | | JJA | 32.3 | 34.4 | 45.7 | 6.0 | 41.4 | 24.4 | 47.7 | 0.45 | 0.21 | 54 |
| | | SON | 18.3 | 22.9 | 37.6 | 27.9 | 115.8 | 41.5 | 121.7 | 0.67 | 0.24 | 54 |
| | | DJF | 17.1 | 18.0 | 36.4 | 7.4 | 118.4 | 37.1 | 126.8 | 0.77 | 0.27 | 54 |

**Table A2.** Same as Table A1 but for $NO_2$.

| Type | Scale | Season | OBS | $MOD_C$ | $MOD_M$ | $nMB_C$ | $nMB_M$ | $nRMSE_C$ | $nRMSE_M$ | $PCC_C$ | $PCC_M$ | N |
|------|-------|--------|-----|---------|---------|---------|---------|-----------|-----------|---------|---------|---|
| RUR | Daily | MAM | 5.0 | 6.7 | — | 35.0 | — | 83.6 | — | 0.56 | — | $0.53\ 10^6$ |
| | | JJA | 3.7 | 5.2 | — | 41.6 | — | 92.4 | — | 0.45 | — | $0.51\ 10^6$ |
| | | SON | 5.6 | 7.1 | — | 27.2 | — | 75.2 | — | 0.59 | — | $0.53\ 10^6$ |
| | | DJF | 7.5 | 8.4 | — | 12.1 | — | 70.4 | — | 0.60 | — | $0.53\ 10^6$ |
| RUR | Monthly | MAM | 4.2 | 6.6 | — | 49.2 | — | 74.3 | — | 0.46 | — | 54 |
| | | JJA | 3.1 | 5.0 | — | 55.7 | — | 78.6 | — | 0.39 | — | 54 |
| | | SON | 4.9 | 7.2 | — | 38.9 | — | 67.2 | — | 0.49 | — | 54 |
| | | DJF | 6.4 | 8.5 | — | 22.4 | — | 58.7 | — | 0.57 | — | 54 |
| URB | Daily | MAM | 10.4 | 6.5 | — | -38.0 | — | 66.0 | — | 0.53 | — | $1.39\ 10^6$ |
| | | JJA | 7.8 | 5.2 | — | -33.7 | — | 66.5 | — | 0.41 | — | $1.38\ 10^6$ |
| | | SON | 11.5 | 6.8 | — | -40.9 | — | 64.8 | — | 0.51 | — | $1.38\ 10^6$ |
| | | DJF | 14.6 | 8.1 | — | -44.8 | — | 66.4 | — | 0.57 | — | $1.37\ 10^6$ |
| URB | Monthly | MAM | 9.8 | 6.3 | — | -36.2 | — | 52.1 | — | 0.54 | — | 54 |
| | | JJA | 7.1 | 4.9 | — | -32.0 | — | 50.1 | — | 0.49 | — | 54 |
| | | SON | 10.8 | 6.7 | — | -39.1 | — | 51.7 | — | 0.55 | — | 54 |
| | | DJF | 13.8 | 8.2 | — | -43.2 | — | 54.8 | — | 0.61 | — | 54 |

**Table A3.** Same as Table A1 but for CO.

| Type | Scale | Season | OBS | $MOD_C$ | $MOD_M$ | $nMB_C$ | $nMB_M$ | $nRMSE_C$ | $nRMSE_M$ | $PCC_C$ | $PCC_M$ | N |
|------|-------|--------|-----|---------|---------|---------|---------|-----------|-----------|---------|---------|---|
| RUR | Daily | MAM | 208.5 | 199.5 | 120.3 | -4.3 | -42.3 | 82.6 | 91.5 | 0.18 | 0.19 | $0.04\ 10^6$ |
| | | JJA | 169.1 | 145.3 | 116.0 | -14.1 | -31.4 | 84.8 | 88.4 | 0.14 | 0.16 | $0.04\ 10^6$ |
| | | SON | 208.4 | 178.6 | 124.9 | -14.3 | -40.1 | 86.1 | 93.6 | 0.21 | 0.20 | $0.04\ 10^6$ |
| | | DJF | 273.4 | 232.8 | 134.4 | -14.8 | -50.8 | 82.8 | 96.4 | 0.27 | 0.22 | $0.04\ 10^6$ |
| RUR | Monthly | MAM | 179.7 | 196.1 | 114.4 | 9.2 | -34.2 | 32.0 | 41.4 | 0.46 | 0.51 | 54 |
| | | JJA | 138.5 | 145.8 | 111.7 | 2.7 | -19.7 | 30.7 | 33.6 | 0.53 | 0.56 | 54 |
| | | SON | 173.9 | 177.4 | 119.1 | 0.2 | -29.0 | 34.0 | 40.3 | 0.53 | 0.57 | 54 |
| | | DJF | 229.4 | 227.7 | 129.2 | -4.0 | -40.5 | 38.5 | 50.1 | 0.58 | 0.56 | 54 |
| URB | Daily | MAM | 308.4 | 197.4 | 120.5 | -36.0 | -60.9 | 71.1 | 88.5 | 0.35 | 0.19 | $0.28\ 10^6$ |
| | | JJA | 234.0 | 148.2 | 118.7 | -36.7 | -49.3 | 79.5 | 85.7 | 0.15 | 0.10 | $0.27\ 10^6$ |
| | | SON | 351.8 | 182.2 | 126.2 | -48.2 | -64.1 | 88.4 | 100.7 | 0.35 | 0.16 | $0.28\ 10^6$ |
| | | DJF | 498.4 | 232.8 | 137.3 | -53.3 | -72.5 | 94.4 | 109.0 | 0.33 | 0.13 | $0.29\ 10^6$ |
| URB | Monthly | MAM | 277.1 | 193.2 | 115.8 | -27.9 | -55.9 | 44.3 | 64.1 | 0.50 | 0.34 | 54 |
| | | JJA | 206.8 | 146.7 | 116.0 | -29.3 | -43.0 | 44.7 | 53.6 | 0.47 | 0.42 | 54 |
| | | SON | 309.8 | 180.4 | 121.6 | -40.6 | -58.7 | 53.7 | 68.1 | 0.56 | 0.38 | 54 |
| | | DJF | 425.1 | 227.9 | 133.2 | -45.8 | -68.0 | 58.2 | 77.3 | 0.59 | 0.37 | 54 |

**Table A4.** Same as Table A1 but for $SO_2$.

| Type | Scale | Season | OBS | $MOD_C$ | $MOD_M$ | $nMB_C$ | $nMB_M$ | $nRMSE_C$ | $nRMSE_M$ | $PCC_C$ | $PCC_M$ | N |
|------|-------|--------|-----|---------|---------|---------|---------|-----------|-----------|---------|---------|---|
| RUR | Daily | MAM | 1.5 | 1.6 | 1.9 | 7.4 | 28.0 | 124.1 | 118.7 | 0.35 | 0.40 | $0.20 \cdot 10^6$ |
|  |  | JJA | 1.3 | 1.6 | 1.7 | 23.6 | 32.8 | 153.2 | 140.2 | 0.26 | 0.27 | $0.18 \cdot 10^6$ |
|  |  | SON | 1.5 | 1.9 | 2.3 | 26.8 | 56.9 | 149.0 | 153.5 | 0.33 | 0.31 | $0.19 \cdot 10^6$ |
|  |  | DJF | 2.0 | 1.8 | 2.8 | -8.7 | 40.1 | 140.3 | 150.0 | 0.36 | 0.35 | $0.21 \cdot 10^6$ |
| RUR | Monthly | MAM | 1.3 | 1.2 | 1.9 | -3.5 | 57.2 | 66.5 | 91.7 | 0.32 | 0.36 | 54 |
|  |  | JJA | 1.1 | 1.2 | 1.7 | 6.5 | 61.7 | 69.4 | 91.7 | 0.27 | 0.29 | 54 |
|  |  | SON | 1.2 | 1.3 | 2.3 | 11.4 | 93.8 | 72.3 | 126.6 | 0.27 | 0.28 | 54 |
|  |  | DJF | 1.5 | 1.3 | 2.8 | -12.4 | 85.8 | 70.9 | 123.4 | 0.28 | 0.30 | 54 |
| URB | Daily | MAM | 2.9 | 2.1 | 2.0 | -27.1 | -28.4 | 228.0 | 230.6 | 0.16 | 0.07 | $0.60 \cdot 10^6$ |
|  |  | JJA | 2.1 | 2.1 | 1.9 | 3.1 | -7.5 | 218.2 | 216.5 | 0.17 | 0.06 | $0.52 \cdot 10^6$ |
|  |  | SON | 2.8 | 2.4 | 2.5 | -12.7 | -11.6 | 245.4 | 249.0 | 0.18 | 0.05 | $0.57 \cdot 10^6$ |
|  |  | DJF | 4.7 | 2.3 | 2.9 | -50.9 | -38.4 | 238.4 | 243.4 | 0.21 | 0.05 | $0.65 \cdot 10^6$ |
| URB | Monthly | MAM | 1.8 | 1.4 | 1.9 | -18.6 | 4.8 | 65.4 | 75.7 | 0.29 | 0.29 | 54 |
|  |  | JJA | 1.5 | 1.4 | 1.7 | -3.8 | 17.0 | 68.4 | 78.6 | 0.21 | 0.21 | 54 |
|  |  | SON | 1.7 | 1.6 | 2.2 | -8.0 | 26.2 | 67.0 | 86.1 | 0.26 | 0.24 | 54 |
|  |  | DJF | 2.3 | 1.6 | 2.6 | -35.7 | 9.1 | 68.0 | 84.3 | 0.32 | 0.27 | 54 |

**Table A5.** Same as Table A1 but for $PM_{10}$. Aerosol concentrations are expressed in $\mu g\,m^{-3}$.

| Type | Scale | Season | OBS | $MOD_C$ | $MOD_M$ | $nMB_C$ | $nMB_M$ | $nRMSE_C$ | $nRMSE_M$ | $PCC_C$ | $PCC_M$ | N |
|------|-------|--------|-----|---------|---------|---------|---------|-----------|-----------|---------|---------|---|
| RUR | Daily | MAM | 18.8 | 24.5 | 25.8 | 30.2 | 37.2 | 79.7 | 127.0 | 0.51 | 0.31 | $0.46\ 10^6$ |
| | | JJA | 16.6 | 19.8 | 23.3 | 19.6 | 40.4 | 84.0 | 103.7 | 0.33 | 0.40 | $0.46\ 10^6$ |
| | | SON | 17.6 | 18.5 | 22.7 | 5.0 | 28.7 | 77.5 | 122.5 | 0.44 | 0.23 | $0.46\ 10^6$ |
| | | DJF | 20.4 | 20.7 | 22.8 | 1.3 | 11.5 | 82.9 | 149.0 | 0.51 | 0.10 | $0.44\ 10^6$ |
| RUR | Monthly | MAM | 17.9 | 24.4 | 23.4 | 35.9 | 37.7 | 65.1 | 87.3 | 0.54 | 0.35 | 54 |
| | | JJA | 15.7 | 20.0 | 20.4 | 25.4 | 37.8 | 56.7 | 70.5 | 0.48 | 0.41 | 54 |
| | | SON | 16.4 | 17.9 | 19.9 | 7.1 | 30.4 | 53.3 | 85.3 | 0.53 | 0.26 | 54 |
| | | DJF | 18.3 | 19.7 | 18.7 | 5.3 | 23.7 | 62.9 | 104.5 | 0.48 | 0.11 | 54 |
| URB | Daily | MAM | 25.9 | 25.1 | 27.3 | -3.0 | 5.7 | 71.8 | 106.2 | 0.43 | 0.32 | $1.48\ 10^6$ |
| | | JJA | 21.1 | 20.1 | 24.0 | -4.6 | 14.0 | 71.0 | 82.7 | 0.30 | 0.41 | $1.46\ 10^6$ |
| | | SON | 26.4 | 19.0 | 23.7 | -28.0 | -10.3 | 82.8 | 104.1 | 0.37 | 0.19 | $1.46\ 10^6$ |
| | | DJF | 33.6 | 21.2 | 23.8 | -36.8 | -29.2 | 95.5 | 125.8 | 0.39 | 0.07 | $1.45\ 10^6$ |
| URB | Monthly | MAM | 22.9 | 24.4 | 24.5 | 7.7 | 8.1 | 52.8 | 72.0 | 0.51 | 0.31 | 54 |
| | | JJA | 18.8 | 19.9 | 20.8 | 6.8 | 15.1 | 48.3 | 55.4 | 0.45 | 0.39 | 54 |
| | | SON | 22.1 | 18.1 | 20.9 | -18.7 | -7.3 | 49.2 | 68.5 | 0.54 | 0.21 | 54 |
| | | DJF | 27.2 | 19.9 | 20.1 | -26.5 | -27.7 | 58.3 | 85.0 | 0.53 | 0.03 | 54 |

**Table A6.** Same as Table A5 but for $PM_{2.5}$.

| Type | Scale | Season | OBS | $MOD_C$ | $MOD_M$ | $nMB_C$ | $nMB_M$ | $nRMSE_C$ | $nRMSE_M$ | $PCC_C$ | $PCC_M$ | N |
|------|-------|--------|-----|---------|---------|---------|---------|-----------|-----------|---------|---------|---|
| RUR | Daily | MAM | 11.9 | 15.8 | 11.7 | 32.6 | -1.7 | 88.8 | 83.8 | 0.51 | 0.40 | $0.19\ 10^6$ |
| | | JJA | 9.6 | 13.1 | 11.2 | 36.6 | 17.1 | 106.9 | 82.7 | 0.34 | 0.37 | $0.19\ 10^6$ |
| | | SON | 11.2 | 12.1 | 10.4 | 7.7 | -7.5 | 95.6 | 92.7 | 0.41 | 0.33 | $0.19\ 10^6$ |
| | | DJF | 14.8 | 13.1 | 9.7 | -11.1 | -34.3 | 93.0 | 110.8 | 0.52 | 0.26 | $0.18\ 10^6$ |
| RUR | Monthly | MAM | 11.0 | 15.4 | 11.5 | 41.3 | 6.2 | 73.6 | 58.2 | 0.53 | 0.51 | 54 |
| | | JJA | 9.0 | 13.0 | 10.9 | 40.9 | 20.9 | 68.6 | 50.1 | 0.50 | 0.58 | 54 |
| | | SON | 9.6 | 11.3 | 10.0 | 16.0 | 3.7 | 60.9 | 60.4 | 0.53 | 0.48 | 54 |
| | | DJF | 11.7 | 11.9 | 9.4 | 0.9 | -17.2 | 67.9 | 73.8 | 0.48 | 0.36 | 54 |
| URB | Daily | MAM | 15.1 | 16.1 | 12.2 | 6.7 | -19.4 | 78.4 | 82.4 | 0.45 | 0.36 | $0.59\ 10^6$ |
| | | JJA | 11.0 | 13.3 | 11.4 | 21.2 | 3.4 | 69.7 | 61.4 | 0.42 | 0.41 | $0.59\ 10^6$ |
| | | SON | 15.2 | 12.1 | 10.7 | -20.4 | -30.1 | 77.2 | 85.1 | 0.44 | 0.30 | $0.59\ 10^6$ |
| | | DJF | 22.2 | 13.4 | 10.3 | -39.7 | -53.3 | 91.8 | 107.4 | 0.47 | 0.19 | $0.58\ 10^6$ |
| URB | Monthly | MAM | 14.1 | 15.8 | 12.0 | 14.8 | -12.6 | 57.3 | 54.4 | 0.54 | 0.49 | 54 |
| | | JJA | 10.6 | 13.2 | 11.2 | 25.0 | 5.5 | 54.4 | 42.4 | 0.49 | 0.54 | 54 |
| | | SON | 13.3 | 11.5 | 10.3 | -13.1 | -21.4 | 52.9 | 57.4 | 0.56 | 0.43 | 54 |
| | | DJF | 18.2 | 12.4 | 9.9 | -31.5 | -45.4 | 64.2 | 76.7 | 0.53 | 0.30 | 54 |

## Appendix B: Urban background stations

The statistics found in Table 4 and in Table 5 are presented here for the subset of urban background stations.

**Table B1.** Same as Table 4 but for urban background stations.

| Scale | Pollutant | OBS | $MOD_C$ | $MOD_M$ | $nMB_C$ | $nMB_M$ | $nRMSE_C$ | $nRMSE_M$ | $PCC_C$ | $PCC_M$ | N |
|-------|-----------|-----|---------|---------|---------|---------|-----------|-----------|---------|---------|---|
| Daily | $O_3$ | 25.3 | 27.3 | 41.5 | 8.0 | 64.1 | 34.3 | 75.2 | 0.72 | 0.54 | $5.13\ 10^6$ |
| | $NO_2$ | 11.1 | 6.6 | — | -40.2 | — | 67.5 | — | 0.56 | — | $5.52\ 10^6$ |
| | CO | 350.8 | 191.0 | 125.8 | -45.6 | -64.1 | 91.0 | 105.2 | 0.39 | 0.19 | $1.13\ 10^6$ |
| | $SO_2$ | 3.2 | 2.2 | 2.3 | -29.3 | -25.8 | 246.8 | 250.8 | 0.18 | 0.08 | $2.34\ 10^6$ |
| | $PM_{10}$ | 26.7 | 21.4 | 24.7 | -20.0 | -7.5 | 85.1 | 112.2 | 0.36 | 0.19 | $5.84\ 10^6$ |
| | $PM_{2.5}$ | 15.8 | 13.7 | 11.1 | -13.3 | -29.6 | 86.2 | 96.1 | 0.41 | 0.24 | $2.35\ 10^6$ |
| Monthly | $O_3$ | 24.8 | 26.6 | 41.6 | 10.0 | 81.3 | 31.6 | 87.6 | 0.61 | 0.22 | 216 |
| | $NO_2$ | 10.4 | 6.5 | — | -37.6 | — | 52.2 | — | 0.54 | — | 216 |
| | CO | 307.7 | 188.1 | 121.9 | -36.2 | -56.8 | 50.5 | 66.1 | 0.53 | 0.38 | 216 |
| | $SO_2$ | 1.9 | 1.5 | 2.1 | -17.8 | 13.8 | 67.2 | 81.3 | 0.28 | 0.25 | 216 |
| | $PM_{10}$ | 22.7 | 20.6 | 21.5 | -7.6 | -2.9 | 52.1 | 70.2 | 0.51 | 0.24 | 216 |
| | $PM_{2.5}$ | 14.0 | 13.2 | 10.9 | -1.1 | -18.3 | 57.1 | 57.7 | 0.53 | 0.44 | 216 |

**Table B2.** Same as Table 5 but for urban background stations.

| Pollutant | $b_O$ | $\epsilon_{O-}$ | $\epsilon_{O+}$ | $b_C$ | $\epsilon_{C-}$ | $\epsilon_{C+}$ | $b_M$ | $\epsilon_{M-}$ | $\epsilon_{M+}$ |
|-----------|-------|-----------------|-----------------|-------|-----------------|-----------------|-------|-----------------|-----------------|
| $O_3$ | **+0.12** (+0.49 %/yr) | -0.17 | +0.33 | **+0.24** (+0.92 %/yr) | +0.02 | +0.47 | -0.06 (-0.13 %/yr) | -0.23 | +0.11 |
| $NO_2$ | **-0.25** (-2.3 %/yr) | -0.36 | -0.17 | **-0.17** (-2.6 %/yr) | -0.23 | -0.13 | — | — | — |
| CO | **-5.85** (-1.9 %/yr) | -8.82 | -2.72 | **-4.19** (-2.2 %/yr) | -6.00 | -3.09 | **-0.72** (-0.59 %/yr) | -0.98 | -0.27 |
| $SO_2$ | **-0.040** (-2.1 %/yr) | -0.051 | -0.029 | **-0.070** (-4.6 %/yr) | -0.074 | -0.064 | **-0.031** (-1.5 %/yr) | -0.046 | -0.015 |
| $PM_{10}$ | **-0.38** (-1.7 %/yr) | -0.52 | -0.23 | **-0.68** (-3.2 %/yr) | -0.82 | -0.59 | -0.05 (-0.24 %/yr) | -0.20 | +0.034 |
| $PM_{2.5}$ | **-0.23** (-1.6 %/yr) | -0.35 | -0.13 | **-0.53** (-3.5 %/yr) | -0.65 | -0.47 | -0.04 (-0.33 %/yr) | -0.11 | +0.01 |

## Appendix C: Trends

Given our monthly time series does not contain tied or missing values, the Seasonal Mann-Kendall statistic, $S'$, and its variance, $Var[S']$, can be obtained as follows:

$$S' = \sum_{g=1}^{m} S_g = \sum_{g=1}^{m} \sum_{i=1}^{n-1} \sum_{j=i+1}^{n} sgn(x_{jg} - x_{ig}) \tag{C1a}$$

$$Var[S'] = \sum_{g=1}^{m} \sigma_g^2 + \sum_{g,h} \sigma_{gh} = \frac{1}{18}[n(n-1)(2n+5)] + \frac{1}{3}[K_{gh} + 4\sum_{j=1}^{n} R_{jg}R_{jh} - n(n+1)^2] \tag{C1b}$$

$$K_{gh} = \sum_{i=1}^{n-1} \sum_{j=1}^{n} sgn[(x_{jg} - x_{ig})(x_{jh} - x_{ih})] \tag{C1c}$$

Where $n$ and $m$ are the number of years and seasons (i.e. here monthly values), respectively, $S_g$ is the Mann-Kendall statistic for each $g_{th}$ season, $R_g$ and $R_h$ are Spearman's correlation coefficients for seasons $g$ and $h$, respectively, and $sgn(x)$ is the sign function. Seasonal Theil-Sen slopes (i.e. annual trends) are then derived from $S'$ (Hussain and Mahmud (2019); Hipel and McLeod (1994); Hirsch and Slack (1984)). The confidence intervals, derived from $Var[S']$, are computed accounting for seasonality but not for autocorrelation, mainly due to the detection of a potential bug in the function *correlated_multivariate_test* from the *Python* library *pyMannKendall* (Hussain and Mahmud, 2019), which at the date of this work's submission remained unresolved.

## Appendix D: QA flags

Using the metadata available in GHOST, a quality assurance screening is applied by removing all air quality observations associated with a set of flags detailed in Table D1. In addition, we detected a few very low CO concentrations in specific regions during specific time periods, which we suspect originate from errors of units when the Member State reported its observations to the EEA. Therefore, as a precautionary measure, all CO hourly observations below 1 ppbv were discarded in this study.

**Table D1.** Description of the GHOST quality-assurance flags used on the EEA air quality observational dataset.

| Flag | Description |
| --- | --- |
| 0 | Measurement is missing (i.e. —). |
| 1 | Value is infinite – occurs when data values are outside of the range that *float32* data type can handle (-3.4E+38 to +3.4E+38). |
| 2 | Measurement is negative in absolute terms. |
| 3 | Measurement is equal to zero. |
| 6 | Measurements are associated with data quality flags given by the data provider which have been decreed by the GHOST project architects as being associated with substantial uncertainty/bias. |
| 8 | After screening by key QA flags, no valid data remains to average in the temporal window. |
| 10 | The measurement methodology used has not yet been mapped to standardised dictionaries of measurement methodologies. |
| 18 | The specific name of the measurement method is unknown. |
| 20 | The primary sampling is not appropriate to prepare the specific parameter for subsequent measurement. |
| 21 | The sample preparation is not appropriate to prepare the specific parameter for subsequent measurement. |
| 22 | The measurement methodology used is not known to be able to measure the specific parameter. |
| 23 | The specific measurement methodology has been decreed not to conform to QA standards as the method is not sufficiently proven/ subject to substantial biases/uncertainty. |
| 72 | Measurement is below or equal to the preferential lower limit of detection. |
| 75 | Measurement is above or equal to the preferential upper limit of detection. |
| 82 | The preferential resolution for the measurement is coarser than a set limit (variable by measured parameter). |
| 83 | The resolution of the measurement is analysed month by month. If the minimum difference between observations is coarser than a set limit (variable by measured parameter), measurements are flagged. |
| 90 | Check for persistently recurring values. Check is done by using a moving window of 9 measurements. If 5/6 (i.e. 83.33%) of values in the window are the same then the entire window is flagged. |
| 91 | Check for persistently recurring values. Check is done by using a moving window of 12 measurements. If 9/12 (i.e. 75%) of values in the window are the same, then the entire window is flagged. |
| 92 | Check for persistently recurring values. Check is done by using a moving window of 24 measurements. If 16/24 (i.e. 66.66%) of values in the window are the same, then the entire window is flagged. |
| 110 | The measured value is below or greater than scientifically feasible lower/upper limits (400, 600, 30000 and 3000 ppbv for $O_3$, $NO_2$, CO and $SO_2$, and 50000 $\mu g\,m^{-3}$ for $PM_{10}$ and $PM_{2.5}$). |
| 111 | The median of the measurements in a month is greater than a scientifically feasible limit (120, 200, 7500 and 750 ppbv for $O_3$, $NO_2$, CO and $SO_2$, and 5000 $\mu g\,m^{-3}$ for $PM_{10}$ and $PM_{2.5}$). |
| 112 | Data has been reported to be an outlier through data flags by the network data reporters (and not manually checked and verified as valid). |
| 113 | Data has been found and decreed manually to be an outlier. |
| 131 | 2 out of 3 months' distributions are classed as Zone 6 or higher, suggesting there are potentially systematic reasons for the inconsistent distributions across the 3 months. |
| 132 | 4 out of 6 months' distributions are classed as Zone 6 or higher, suggesting there are potentially systematic reasons for the inconsistent distributions across the 6 months. |
| 133 | 8 out of 12 months' distributions are classed as Zone 6 or higher, suggesting there are potentially systematic reasons for the inconsistent distributions across the 12 months. |

*Author contributions.* AL carried out the analysis. AL and HP contributed to the conception and design of the study. DB was responsible for the acquisition and preprocessing of the air quality data through the GHOST project. AL, HP, ZC, RFMT, HA, CPGP, OJ, AS and JB
contributed to the interpretation of results. AL and HP were responsible for writing the article, with a review from CPGP and AS.

*Competing interests.* The authors declare that they have no conflict of interest.

*Acknowledgements.* This research has received funding from the European Research Council (ERC), in the frame of the EARLY-ADAPT project (https://early-adapt.eu/), under the European Union's Horizon 2020 research and innovation programme (Grant agreement No. 865564), as well as the MITIGATE project (PID2020-116324RA-I00 / AEI / 10.13039/501100011033) from the Agencia Estatal de Investi-
gacion (AEI). We also acknowledge support by the the AXA Research Fund and Red Temática ACTRIS España (CGL2017-90884-REDT), PRACE and RES for awarding us access to MareNostrum Supercomputer in the Barcelona Supercomputing Center, and H2020 ACTRIS IMP (#871115).

## Appendix: References

Aldabe, J., Elustondo, D., Santamaría, C., Lasheras, E., Pandolfi, M., Alastuey, A., Querol, X., and Santamaría, J. M.: Chemical characteri-
sation and source apportionment of PM2.5 and PM10 at rural, urban and traffic sites in Navarra (North of Spain), Atmospheric Research, 102, 191–205, https://doi.org/10.1016/j.atmosres.2011.07.003, 2011.

Ali, M. A., Bilal, M., Wang, Y., Nichol, J. E., Mhawish, A., Qiu, Z., de Leeuw, G., Zhang, Y., Zhan, Y., Liao, K., Almazroui, M., Dambul, R., Shahid, S., and Islam, M. N.: Accuracy assessment of CAMS and MERRA-2 reanalysis PM2.5 and PM10 concentrations over China, Atmospheric Environment, 288, 119 297, https://doi.org/10.1016/j.atmosenv.2022.119297, 2022.

Barré, J., Petetin, H., Colette, A., Guevara, M., Peuch, V.-H., Rouil, L., Engelen, R., Inness, A., Flemming, J., Pérez García-Pando, C., Bowdalo, D., Meleux, F., Geels, C., Christensen, J. H., Gauss, M., Benedictow, A., Tsyro, S., Friese, E., Struzewska, J., Kaminski, J. W., Douros, J., Timmermans, R., Robertson, L., Adani, M., Jorba, O., Joly, M., and Kouznetsov, R.: Estimating lockdown-induced European NO2 changes using satellite and surface observations and air quality models, Atmospheric Chemistry and Physics, 21, 7373–7394, https://doi.org/10.5194/acp-21-7373-2021, 2021.

Bauwens, M., Compernolle, S., Stavrakou, T., Müller, J. F., van Gent, J., Eskes, H., Levelt, P. F., van der A, R., Veefkind, J. P., Vlietinck, J., Yu, H., and Zehner, C.: Impact of Coronavirus Outbreak on NO2 Pollution Assessed Using TROPOMI and OMI Observations, Geophysical Research Letters, 47, 1–9, https://doi.org/10.1029/2020GL087978, 2020.

Bosilovich, M., Akella, S., Coy, L., Cullather, R., Draper, C., Gelaro, R., Kovach, R., Liu, Q., Molod, A., Norris, P., Wargan, K., Chao, W., Reichle, R., Takacs, L., Vikhliaev, Y., Bloom, S., Collow, A., Firth, S., Labow, G., Partyka, G., Pawson, S., Reale, O., Schubert, S. D., and
Suarez, M.: MERRA-2 : Initial Evaluation of the Climate, NASA Technical Report Series on Global Modeling and Data Assimilation, 43, 139, 2015.

Buchard, V., da Silva, A. M., Randles, C. A., Colarco, P., Ferrare, R., Hair, J., Hostetler, C., Tackett, J., and Winker, D.: Evaluation of the surface PM2.5 in Version 1 of the NASA MERRA Aerosol Reanalysis over the United States, Atmospheric Environment, 125, 100–111, https://doi.org/10.1016/j.atmosenv.2015.11.004, 2016.

Buchard, V., Randles, C. A., da Silva, A. M., Darmenov, A., Colarco, P. R., Govindaraju, R., Ferrare, R., Hair, J., Beyersdorf, A. J., Ziemba, L. D., and Yu, H.: The MERRA-2 aerosol reanalysis, 1980 onward. Part II: Evaluation and case studies, Journal of Climate, 30, 6851–6872, https://doi.org/10.1175/JCLI-D-16-0613.1, 2017a.

Buchard, V., Randles, C. A., da Silva, A. M., Darmenov, A., Colarco, P. R., Govindaraju, R., Ferrare, R., Hair, J., Beyersdorf, A. J., Ziemba, L. D., and Yu, H.: The MERRA-2 aerosol reanalysis, 1980 onward. Part II: Evaluation and case studies, Journal of Climate, 30, 6851–6872,
https://doi.org/10.1175/JCLI-D-16-0613.1, 2017b.

Chin, M., Ginoux, P., Kinne, S., Torres, O., Holben, B. N., Duncan, B. N., Martin, R. V., Logan, J. A., Higurashi, A., and Nakajima, T.: Tropospheric aerosol optical thickness from the GOCART model and comparisons with satellite and sun photometer measurements, Journal of the Atmospheric Sciences, 59, 461–483, https://doi.org/10.1175/1520-0469(2002)059<0461:taotft>2.0.co;2, 2002.

Colarco, P., Da Silva, A., Chin, M., and Diehl, T.: Online simulations of global aerosol distributions in the NASA GEOS-4
model and comparisons to satellite and ground-based aerosol optical depth, Journal of Geophysical Research Atmospheres, 115, https://doi.org/10.1029/2009JD012820, 2010.

Cuesta, J., Eremenko, M., Liu, X., Dufour, G., Cai, Z., Höpfner, M., Von Clarmann, T., Sellitto, P., Foret, G., Gaubert, B., Beekmann, M., Orphal, J., Chance, K., Spurr, R., and Flaud, J. M.: Satellite observation of lowermost tropospheric ozone by multispectral synergism of IASI thermal infrared and GOME-2 ultraviolet measurements over Europe, Atmospheric Chemistry and Physics, 13, 9675–9693,
https://doi.org/10.5194/acp-13-9675-2013, 2013.

Darmenov, A. and da Silva, A.: The Quick Fire Emissions Dataset (QFED) - Documentation of versions 2.1, 2.2 and 2.4., NASA Technical Report Series on Global Modeling and Data Assimilation, 32, 2013.

Dee, D. and Uppala, S.: Variational bias correction in ERA-Interim, p. 26, http://rda.ucar.edu/datasets/ds627.1/docs/IFS_documentation/ifs. 2b.variational_bias_correction.pdf, 2008.

Diehl, T., Heil, A., Chin, M., Pan, X., Streets, D., Schultz, M., and Kinne, S.: Anthropogenic, biomass burning, and volcanic emissions of black carbon, organic carbon, and $SO_2$ from 1980 to 2010 for hindcast model experiments, Atmospheric Chemistry and Physics Discussions, 12, 24895–24954, https://doi.org/10.5194/acpd-12-24895-2012, 2012.

Duncan, B. N., Martin, R. V., Staudt, A. C., Yevich, R., and Logan, J. A.: Interannual and seasonal variability of biomass burning emissions constrained by satellite observations, Journal of Geophysical Research: Atmospheres, 108, https://doi.org/10.1029/2002jd002378, 2003.

EEA:   AirBase   -   The   European   air   quality   database,   https://www.eea.europa.eu/data-and-maps/data/ airbase-the-european-air-quality-database-8, 2014.

EEA: Air Quality e-Reporting (AQ e-Reporting), https://www.eea.europa.eu/data-and-maps/data/aqereporting-8, 2018.

European Environment Agency: European Union emission inventory report 1990-2019, EEA Report No 05/2021, Tech. Rep. 6, https://www. eea.europa.eu/publications/european-union-emission-inventory-report-1, 2021.

Flemming, J., Huijnen, V., Arteta, J., Bechtold, P., Beljaars, A., Blechschmidt, A. M., Diamantakis, M., Engelen, R. J., Gaudel, A., Inness, A., Jones, L., Josse, B., Katragkou, E., Marecal, V., Peuch, V. H., Richter, A., Schultz, M. G., Stein, O., and Tsikerdekis, A.: Tropospheric chemistry in the integrated forecasting system of ECMWF, Geoscientific Model Development, 8, 975–1003, https://doi.org/10.5194/gmd-8-975-2015, 2015.

Gelaro, R., McCarty, W., Suárez, M. J., Todling, R., Molod, A., Takacs, L., Randles, C. A., Darmenov, A., Bosilovich, M. G., Reichle, R.,
Wargan, K., Coy, L., Cullather, R., Draper, C., Akella, S., Buchard, V., Conaty, A., da Silva, A. M., Gu, W., Kim, G. K., Koster, R., Lucchesi, R., Merkova, D., Nielsen, J. E., Partyka, G., Pawson, S., Putman, W., Rienecker, M., Schubert, S. D., Sienkiewicz, M., and Zhao, B.: The modern-era retrospective analysis for research and applications, version 2 (MERRA-2), Journal of Climate, 30, 5419–5454, https://doi.org/10.1175/JCLI-D-16-0758.1, 2017.

Granier, C., Bessagnet, B., Bond, T., D'Angiola, A., van der Gon, H. D., Frost, G. J., Heil, A., Kaiser, J. W., Kinne, S., Klimont, Z.,
Kloster, S., Lamarque, J. F., Liousse, C., Masui, T., Meleux, F., Mieville, A., Ohara, T., Raut, J. C., Riahi, K., Schultz, M. G., Smith, S. J., Thompson, A., van Aardenne, J., van der Werf, G. R., and van Vuuren, D. P.: Evolution of anthropogenic and biomass burning emissions of air pollutants at global and regional scales during the 1980–2010 period, Climatic Change 2011 109:1, 109, 163–190, https://doi.org/10.1007/S10584-011-0154-1, 2011.

Guenther, A., Nicholas, C., Fall, R., Klinger, L., Mckay, W. A., and Scholes, B.: A global model of natural volatile organic compound
emissions s Raja the balance Triangle changes in the atmospheric accumulation rates of greenhouse Triangle Several inventories of natural and Exposure Assessment global scales have been two classes Fores, J. Geophys. Res., 100, 8873–8892, 1995.

Hersbach, H., Bell, B., Berrisford, P., Hirahara, S., Horányi, A., Muñoz-Sabater, J., Nicolas, J., Peubey, C., Radu, R., Schepers, D., Simmons, A., Soci, C., Abdalla, S., Abellan, X., Balsamo, G., Bechtold, P., Biavati, G., Bidlot, J., Bonavita, M., De Chiara, G., Dahlgren, P., Dee, D., Diamantakis, M., Dragani, R., Flemming, J., Forbes, R., Fuentes, M., Geer, A., Haimberger, L., Healy, S., Hogan, R. J.,
Hólm, E., Janisková, M., Keeley, S., Laloyaux, P., Lopez, P., Lupu, C., Radnoti, G., de Rosnay, P., Rozum, I., Vamborg, F., Villaume, S., and Thépaut, J. N.: The ERA5 global reanalysis, Quarterly Journal of the Royal Meteorological Society, 146, 1999–2049, https://doi.org/10.1002/qj.3803, 2020.

Hipel, K. W. and McLeod, A. I.: Time series modelling of water resources and environmental systems, https://doi.org/10.1016/0022-1694(95)90010-1, 1994.

Hirsch, R. M. and Slack, J. R.: A Nonparametric Trend Test for Seasonal Data With Serial Dependenc, Water Resources Research, 20, 727–732, 1984.

Huijnen, V., Miyazaki, K., Flemming, J., Inness, A., Sekiya, T., and G. Schultz, M.: An intercomparison of tropospheric ozone reanalysis products from CAMS, CAMS interim, TCR-1, and TCR-2, Geoscientific Model Development, 13, 1513–1544, https://doi.org/10.5194/gmd-13-1513-2020, 2020.

Hussain, M. and Mahmud, I.: pyMannKendall: a python package for non parametric Mann Kendall family of trend tests., Journal of Open Source Software, 4, 1556, https://doi.org/10.21105/joss.01556, 2019.

Inness, A., Ades, M., Agustí-Panareda, A., Barr, J., Benedictow, A., Blechschmidt, A. M., Jose Dominguez, J., Engelen, R., Eskes, H., Flemming, J., Huijnen, V., Jones, L., Kipling, Z., Massart, S., Parrington, M., Peuch, V. H., Razinger, M., Remy, S., Schulz, M., and Suttie,

M.: The CAMS reanalysis of atmospheric composition, Atmospheric Chemistry and Physics, 19, 3515–3556, https://doi.org/10.5194/acp-19-3515-2019, 2019.

Kaiser, J. W., Heil, A., Andreae, M. O., Benedetti, A., Chubarova, N., Jones, L., Morcrette, J. J., Razinger, M., Schultz, M. G., Suttie, M., and Van Der Werf, G. R.: Biomass burning emissions estimated with a global fire assimilation system based on observed fire radiative power, Biogeosciences, 9, 527–554, https://doi.org/10.5194/BG-9-527-2012, 2012.

Ma, X., Yan, P., Zhao, T., Jia, X., Jiao, J., Ma, Q., Wu, D., Shu, Z., Sun, X., and Habtemicheal, B. A.: Article evaluations of surface pm10 concentration and chemical compositions in merra-2 aerosol reanalysis over central and eastern china, Remote Sensing, 13, https://doi.org/10.3390/rs13071317, 2021.

Marécal, V., Peuch, V. H., Andersson, C., Andersson, S., Arteta, J., Beekmann, M., Benedictow, A., Bergström, R., Bessagnet, B., Cansado, A., Chéroux, F., Colette, A., Coman, A., Curier, R. L., Van Der Gon, H. A., Drouin, A., Elbern, H., Emili, E., Engelen, R. J., Eskes, H. J., Foret, G., Friese, E., Gauss, M., Giannaros, C., Guth, J., Joly, M., Jaumouillé, E., Josse, B., Kadygrov, N., Kaiser, J. W., Krajsek, K., Kuenen, J., Kumar, U., Liora, N., Lopez, E., Malherbe, L., Martinez, I., Melas, D., Meleux, F., Menut, L., Moinat, P., Morales, T., Parmentier, J., Piacentini, A., Plu, M., Poupkou, A., Queguiner, S., Robertson, L., Rouïl, L., Schaap, M., Segers, A., Sofiev, M., Tarasson, L., Thomas, M., Timmermans, R., Valdebenito, Van Velthoven, P., Van Versendaal, R., Vira, J., and Ung, A.: A regional air quality forecasting system over Europe: The MACC-II daily ensemble production, Geoscientific Model Development, 8, 2777–2813, https://doi.org/10.5194/gmd-8-2777-2015, 2015.

Molod, A., Takacs, L., Suarez, M., Bacmeister, J., Song, I.-S., and Eichmann, A.: NASA / TM – 2012-104606 / Vol 28 Technical Report Series on Global Modeling and Data Assimilation , Volume 28 The GEOS-5 Atmospheric General Circulation Model : Mean Climate and Development from MERRA to Fortuna April 2012, 2012.

Morcrette, J. J., Boucher, O., Jones, L., Salmond, D., Bechtold, P., Beljaars, A., Benedetti, A., Bonet, A., Kaiser, J. W., Razinger, M., Schulz, M., Serrar, S., Simmons, A. J., Sofiev, M., Suttie, M., Tompkins, A. M., and Untch, A.: Aerosol analysis and forecast in the european centre for medium-range weather forecasts integrated forecast system: Forward modeling, Journal of Geophysical Research Atmospheres, 114, 1–17, https://doi.org/10.1029/2008JD011235, 2009.

Navinya, C. D., Vinoj, V., and Pandey, S. K.: Evaluation of pm2.5 surface concentrations simulated by nasa's merra version 2 aerosol reanalysis over india and its relation to the air quality index, Aerosol and Air Quality Research, 20, 1329–1339, https://doi.org/10.4209/aaqr.2019.12.0615, 2020.

Petetin, H., Bowdalo, D., Soret, A., Guevara, M., Jorba, O., Serradell, K., and Pérez García-Pando, C.: Meteorology-normalized impact of the COVID-19 lockdown upon NO2 pollution in Spain, Atmospheric Chemistry and Physics, 20, 11 119–11 141, https://doi.org/10.5194/acp-20-11119-2020, 2020.

Provençal, S., Buchard, V., da Silva, A. M., Leduc, R., and Barrette, N.: Evaluation of PM surface concentrations simulated by Version 1 of NASA's MERRA Aerosol Reanalysis over Europe, Atmospheric Pollution Research, 8, 374–382, https://doi.org/10.1016/j.apr.2016.10.009, 2017a.

Provençal, S., Buchard, V., da Silva, A. M., Leduc, R., Barrette, N., Elhacham, E., and Wang, S. H.: Evaluation of PM2.5 surface concentrations simulated by version 1 of NASA's MERRA aerosol reanalysis over Israel and Taiwan, Aerosol and Air Quality Research, 17, 253–261, https://doi.org/10.4209/aaqr.2016.04.0145, 2017b.

Randerson, J. T., Liu, H., Flanner, M. G., Chambers, S. D., Jin, Y., Hess, P. G., Pfister, G., Mack, M. C., Treseder, K. K., Welp, L. R., Chapin, F. S., Harden, J. W., Goulden, M. L., Lyons, E., Neff, J. C., Schuur, E. A., and Zender, C. S.: The impact of boreal forest fire on climate warming, Science, 314, 1130–1132, https://doi.org/10.1126/science.1132075, 2006.

Randles, C. A., da Silva, A. M., Buchard, V., Colarco, P. R., Darmenov, A., Govindaraju, R., Smirnov, A., Holben, B., Ferrare, R., Hair, J., Shinozuka, Y., and Flynn, C. J.: The MERRA-2 aerosol reanalysis, 1980 onward. Part I: System description and data assimilation evaluation, Journal of Climate, 30, 6823–6850, https://doi.org/10.1175/JCLI-D-16-0609.1, 2017.

Reddy, M. S., Boucher, O., Bellouin, N., Schulz, M., Balkanski, Y., Dufresne, J. L., and Pham, M.: Estimates of global multicomponent aerosol optical depth and direct radiative perturbation in the Laboratoire de Météorologie Dynamique general circulation model, Journal of Geophysical Research D: Atmospheres, 110, 1–16, https://doi.org/10.1029/2004JD004757, 2005.

Rienecker, M., Suarez, M., Todling, R., Bacmeister, J., Takacs, L., Liu, H.-C., Gu, W., Sienkiewicz, M., Koster, R., Gelaro, R., and Nielsen, J.: The GEOS-5 Data Assimilation System-Documentation of Versions 5.0. 1, 5.1. 0, and 5.2. 0, NASA Technical Report, 27, 118 pp., 2008.

Ryu, Y. H. and Min, S. K.: Long-term evaluation of atmospheric composition reanalyses from CAMS, TCR-2, and MERRA-2 over South Korea: Insights into applications, implications, and limitations, Atmospheric Environment, 246, 118 062, https://doi.org/10.1016/j.atmosenv.2020.118062, 2021.

Sindelarova, K., Granier, C., Bouarar, I., Guenther, A., Tilmes, S., Stavrakou, T., Müller, J. F., Kuhn, U., Stefani, P., and Knorr, W.: Global data set of biogenic VOC emissions calculated by the MEGAN model over the last 30 years, Atmospheric Chemistry and Physics, 14, 9317–9341, https://doi.org/10.5194/ACP-14-9317-2014, 2014.

Souri, A. H., Chance, K., Sun, K., Liu, X., and Johnson, M. S.: Dealing with spatial heterogeneity in pointwise-to-gridded-data comparisons, Atmospheric Measurement Techniques, 15, 41–59, https://doi.org/10.5194/amt-15-41-2022, 2022.

Spracklen, D. V., Jimenez, J. L., Carslaw, K. S., Worsnop, D. R., Evans, M. J., Mann, G. W., Zhang, Q., Canagaratna, M. R., Allan, J., Coe, H., McFiggans, G., Rap, A., and Forster, P.: Aerosol mass spectrometer constraint on the global secondary organic aerosol budget, Atmospheric Chemistry and Physics, 11, 12 109–12 136, https://doi.org/10.5194/acp-11-12109-2011, 2011.

Thorsteinsson, T., Jóhannsson, T., Stohl, A., and Kristiansen, N. I.: High levels of particulate matter in Iceland due to direct ash emissions by the Eyjafjallajkull eruption and resuspension of deposited ash, Journal of Geophysical Research: Solid Earth, 117, 1–9, https://doi.org/10.1029/2011JB008756, 2012.

Ukhov, A., Mostamandi, S., Krotkov, N., Flemming, J., da Silva, A., Li, C., Fioletov, V., McLinden, C., Anisimov, A., Alshehri, Y. M., and Stenchikov, G.: Study of SO2 Pollution in the Middle East Using MERRA-2, CAMS Data Assimilation Products, and High-Resolution WRF-Chem Simulations, Journal of Geophysical Research: Atmospheres, 125, https://doi.org/10.1029/2019JD031993, 2020.

Vîrghileanu, M., Săvulescu, I., Mihai, B. A., Nistor, C., and Dobre, R.: Nitrogen dioxide (No2) pollution monitoring with sentinel-5p satellite imagery over europe during the coronavirus pandemic outbreak, Remote Sensing, 12, 1–29, https://doi.org/10.3390/rs12213575, 2020.

Wagner, A., Bennouna, Y., Blechschmidt, A. M., Brasseur, G., Chabrillat, S., Christophe, Y., Errera, Q., Eskes, H., Flemming, J., Hansen, K. M., Inness, A., Kapsomenakis, J., Langerock, B., Richter, A., Sudarchikova, N., Thouret, V., and Zerefos, C.: Comprehensive evaluation of the Copernicus Atmosphere Monitoring Service (CAMS) reanalysis against independent observations: Reactive gases, Elementa, 9, https://doi.org/10.1525/elementa.2020.00171, 2021.