# Peer review of "Long-term evaluation of surface air pollution in CAMSRA and MERRA-2 global reanalyses over Europe (2003-2020)"

_Geoscientific Model Development, 2022_

## Author Comment (AC1)

We thank the anonymous referees for the constructive comments and suggestions they have provided. Throughout this text, referee comments are written in grey, our answers are in black and manuscript modifications are indicated in blue.

**Referee #1**

*General comment*

*The evaluation procedure applied in the paper is sound and provides valuable insight for the user of the data sets. My main concern of the paper is the selection of the stations for the evaluation. Urban stations, which are probably the largest fraction of the European AQ networks, were considered as "background" stations for the paper and combined with the rural stations. I think it is not common practice to include urban stations in the group of background stations. In any case, a stratification between "rural" and "urban" stations is strongly recommended as the scale represented by urban observations is often much smaller as the grid-point resolution of the evaluated models. I think the authors need to quantify the errors against rural stations and urban stations separately, and clarify if the presented results are dominated by the comparison with urban stations or not.*

When background stations are said to be classified into "rural, regional, remote, urban and suburban", this classification refers specifically to the type of area where the background station is located. This typology has been established by the European Environmental Agency (EEA) in their databases AIRBASE and AQ e-Reporting. Thus, a background station located in a urban environment remains unaffected by point emission sources as it is still considered to be representative of the surrounding area.

Nonetheless, all these different types of background stations do not have the same spatial representativeness. Although it depends on many factors (e.g. city morphology, distribution of emission sources, local meteorological conditions), urban and suburban background stations are typically representative of the city (in the case of small cities) or part of the city (in the case of larger ones). Conversely, as they are typically located further from anthropogenic emission sources and in areas with less restricted air flow, rural, rural-regional and rural-remote background stations provide measurements representative of a much larger area. When evaluating a model of relatively coarse spatial resolution, there are intrinsic uncertainties related to these differences of representativeness and the fundamentally different nature of point-based observations and volume-based simulations.

In the initial version of the manuscript submitted to GMD, we initially considered all background stations, thus including rural, rural-remote, rural-regional, urban and suburban background stations and excluding urban traffic and industrial point source stations. This choice was mainly motivated by the fact that we consider gridded observations, meaning that if several background stations are available in a given grid cell, they will be averaged. Although we were assuming that several stations would often be averaged together, thus limiting the issue of the more limited representativeness of urban/suburban background stations compared to rural/regional/remote ones, we realized that a substantial part of these daily gridded observations were actually based on only one single urban/suburban background station.

Therefore, following the comment of the reviewer, we modified our approach and decided to consider only rural-remote, rural-regional and rural background stations, thus excluding urban and suburban background stations, together with the previously excluded traffic and industrial point source stations. This obviously represents a major change of the manuscript, as all tables, figures and discussion are modified in this

revised version of the paper. As we cannot list here all the modifications this represents, we invite the reviewer and the editor to check this revised version. We tried to keep as much as possible of the original structure of the text from the initial version of the manuscript. Additionally, for information purpose, we also provide some brief insights on the evaluation results obtained considering only urban and suburban background stations. To account for these results obtained from the manuscript realignment, a Supplement containing additonal figures has also been prepared.

Although it is explained in the preceding paragraph, we introduced the following modification in the text to avoid any confusion:

L153: "Given the relatively coarse spatial resolution of both reanalyses, only background stations, of greater spatial representativeness, are considered in the evaluation. This includes stations classified as rural, regional, remote, urban or suburban, thus discarding traffic and industrial point source stations. The latter are generally located in areas with restricted/limited air flow and close to emission sources, thus being heavily affected by day-to-day variability in pollution levels." ➜ "Given the relatively coarse spatial resolution of both reanalyses, only rural, rural-regional and rural-remote background stations, of larger spatial representativeness, are considered in the evaluation, thus excluding urban and suburban background stations. Traffic and industrial point source stations have also been discarded, being generally located in areas with limited air flow and close to local emission sources, which causes their pollution concentration levels to be overly driven by day-to-day variability."

*The paper can further be improved by adding more information about the speciation of the PM10 and 2.5 values. It would be interesting to know how large the sulphate (given the differences in SO2) contributions were, and to what extend dust and sea salt contributed to PM in both data sets.*

As this paper focuses on the evaluation of the major pollutants in the two reanalyses, we think that discussing the speciation of particulate matter compounds is beyond the scope of the present study. Such exercise would evidently be of interest if combined with an evaluation against PM speciation observations, but such observations are very scarce for the considered time period (and the number of species included in the study is already quite large).

A series of specific comments are hereunder addressed by the authors:

> ➤ Please include more references in table 1 and discuss their findings.

The suggested references were included in Table 1. Their findings were discussed in the corresponding section for each pollutant and in the brief outline in the Introduction.

> ➤ L75: Add references (links or DOI) of the observational data sets.

The links of both AIRBASE and AQ e-Reporting were added as references to the manuscript.

> ➤ L93: Atmospheric composition fields are only available in grid-point space.

The sentence was modified as follows:

L91: "CAMSRA has a horizontal resolution of approximately 80 km (similar to a regular 0.75º x 0.75º latitude/longitude grid), with data being available both as spectral coefficients (T255) or on a reduced Gaussian grid (N128)." ➔ "CAMSRA has a horizontal resolution of approximately 80 km (similar to a regular 0.75º x 0.75º latitude/longitude grid), with atmospheric composition fields being available only in grid-point space."

➢ L101: SOA is included in the two OM aerosols variables in CAMSRA.

The sentence was clarified:

L101: "Secondary organic aerosols (SOA) of anthropogenic origin are parametrized according to Spracklen et al. (2011)." ➔ "Within OM, secondary organic aerosols (SOA) of anthropogenic origin are parametrized according to Spracklen et al. (2011), based on MACCity CO emissions ."

➢ L120: Please mention here (and also for CAMSRA) the vertical extent of the model level at the surface, which may be important for the interpretation of the results.

The following sentences were modified:

L92: "Its vertical resolution consists of 60 hybrid sigma/pressure model levels, with the top level located at 0.1 hPa." ➔ "Its vertical resolution consists of 60 hybrid sigma/pressure model levels, with the top of the first level at 10 m above ground and the top level located at 0.1 hPa."
L118: "...and has 72 hybrid-eta model levels from the surface to the top at 0.01 hPa. ➔ "...and has 72 hybrid-eta model levels from the surface, with the first level reaching 58 m above ground, to the top at 0.01 hPa."

➢ L125: Please discuss the formulae applied to derive PM10/2.5 for CAMSRA and compare it to the approach used for MERRA-2.

Although the authors do not see what could be commented regarding the differences between CAMSRA and MERRA-2 PM reconstruction, given the different treatment of aerosol species in both reanalyses, we nonetheless added the following information in the CAMSRA section:

L101: "Both PM fields were downloaded directly without any reconstruction or modification, though they are originally reconstructed from the following formulas:

PM10 = RHO  * (SS1/4.3 + SS2/4.3 + DD1 + DD2 + 0.4*DD3 + OM1 + OM2 + SU1 + BC1 + BC2)
PM2.5 = RHO * (SS1/4.3 + 0.5*SS2/4.3 + DD1 + DD2 + 0.7*OM1 + 0.7*OM2 + 0.7*SU1 + BC1 + BC2)

With RHO the air density, SS1/SS2 the sea salt, DD1/DD2/DD3 the desert dust, OM1/OM2 the organic matter, BC1/2 the black carbon, and SU1 the aerosol sulfate mass mixing ratios (with 1/2/3 referring to the aerosol bin, from smallest to largest). The factor 4.3 is applied to convert

the model sea salts expressed at 80% relative humidity in the model (see Reddy et al. (2005)) into dry mass mixing ratios. However, it is worth mentioning that to the best of our knowledge, this correction might need to be revisited in the future to also account for the change of size of the sea salt particles (as mentioned on the CAMS scientific user forum : https://confluence.ecmwf.int/display/CUSF/PM10+and+PM25+global+products, last access: 25th November 2022)."

➢ L152: To consider "urban" stations as "background" stations seems far-fetched. Excluding the urban regime, at least as option seems necessary for the study.

This comment was addressed by the authors under the general comment provided by the anonymous referee.

➢ L153: Provide reference for the classification.

The classification of the ground stations is established by the European Environment Agency (EEA), which classifies them either depending on the predominant emission sources (traffic, industrial, background) or depending on the distribution/density of buildings in the surrounding area (urban, suburban, rural). Both AIRBASE and AQ e-Reporting are databases developed by the EEA, and therefore they follow the mentioned classification.
A reference has been added to the manuscript in L164.

➢ L162: Provide information about the typical grid-box variability, i.e. the deviation (variance) of the observations from the mean.  Please provide information how many of the "observation grid boxes" are actual composed of multi-station means.  For example, a pdf of the number of stations per grid box would be informative.

Since we are now considering only rural, regional and remote background stations, most of the gridded daily observations are now based on one single station. We included some information in the text:

L162: "...more sophisticated methods such as those proposed by Souri et al. (2022) (which employ geostatistical approaches by making use of semivariograms and kriging) might be worth implementing in the future." ➔ "...more sophisticated methods such as those proposed by Souri et al. (2022) (which employ geostatistical approaches by making use of semivariograms and kriging) might be worth implementing in the future. However, when considering only rural, regional and remote background stations, the proportion of gridded daily observations based on one single daily observation (two daily observations) is 96.1% (3.5%) for NO2, 95.4% (4.4%) for O3, 96.7% (3.2%) for SO2, 97.9% (1.9%) for CO, 91.0% (8.5%) for PM10 and 92.5% (7.4%) for PM2.5, these high percentages being explained by the presence of numerous missing values throughout the period of study."

➢ L175: Please, clarify if PCC represents a spatial or a temporal correlation.

The following sentence was added to the manuscript:

L179: "In this study, metrics have been calculated and presented following two different approaches: (1) with a so-called "time-and-space" approach where metrics are calculated in one step, based on all reanalysis-observation pairs available both across the entire domain (or a given country) and over the entire period 2003-2020, or (2) with a so-called "time-then-space" approach where metrics are first calculated at each station, before being combined by taking the median across all stations. In this workframe, "time-and-space" PCC values do not correspond to spatial or temporal correlations but rather to overall spatio-temporal correlations, while "time-then-space" PCC values do correspond to temporal correlations, though spatially averaged."

An example of (1) is shown in Table 4 or in panel b) for each figure, while (2) can be seen in panel a).

➢ L180: Please, clarify the accumulation index I, i.e. is it over time per station, or over stations per time instance).

The following modification was introduced in the text:

L180: "The index $i$ accumulates over time (e.g. daily, monthly) at each station (i.e. gridded cell with available observations). The final value for each statistic is obtained by medianizing across all stations."

➢ Fig. 4: Why is there no PCC for Turkey for MERRA-2.

In this particular case the value of the PCC was negative. It therefore cannot be seen in Fig. 4 as the authors decided to limit the range of the PCC between 0 and 1 in all figures for the sake of a clearer visualization. The following sentence has been added to the caption of Fig. 2, which serves as the caption for all the pollutant figures:
"Only PCC values in the range 0--1 are displayed in b)."

➢ L420: Please, add some information about speciation for PM10 & PM 2.5.

As mentioned before, although it would certainly be interesting to provide some additional insights on the PM speciation, we think it is beyond the scope of the present study as this would require evaluating the model-based PM chemical compounds against in-situ observations. Observational data for speciations is much scarcer compared to reconstitued PM data and extending the analysis to cover speciation would certainly render the manuscript too long.

**Referee #2**

A series of specific and technical comments are hereunder addressed by the authors:

➢ Table 1. The study by Huijnen et al. (GMD 2020) is missing in this list, even though they report to some extent on the performance of surface ozone in the CAMSRA, even with focus over Europe. Is there any reason why it is not needed to include it here, or is it an oversight?

We thank the reviewer for sharing this reference as Huijnen et al. (2020) is indeed an oversight. The study has been included in Table 1 alongside two other additional works.

➢ L85: Please include a reference to the aerosol scheme used in the CAMS reanalysis (Morcrette et al., I believe).

The following sentence was added to the text:

L106: "A detailed description of the aerosol scheme employed in CAMSRA can be found in Morcrette et al. (2009)."

➢ L90: "Meteorological observations and fields are taken from ERA5" while this is probably very close, please note that the CAMS Reanalysis applies its own assimilation of meteorological variables, i.e. it is not identical to ERA5.

The following modification was introduced in the manuscript:

L90: "Meteorological observations and fields are taken from ERA5." ➔ "Meteorological observations are assimilated as in ERA5."

➢ Appendix B: This is useful information, which could be taken over by others in the community. It might be useful if you can additionally better specify flags 110-111: How are these 'scientifically feasible values' defined exactly?

We added this information to the table in Appendix D:

Flag 110: "The measured value is below or greater than scientifically feasible lower/upper limits (400, 600, 30000 and 3000 ppbv for O3, NO2, CO and SO2, and 50000 ug/m3 for PM10 and PM2.5)."
Flag 111: "The median of the measurements in a month is greater than a scientifically feasible limit (120, 200, 7500 and 750 ppbv for O3, NO2, CO and SO2, and 5000 ug/m3 for PM10 and PM2.5)."

A dedicated publication on the GHOST project is currently in preparation (Bowdalo et al.). These upper limits represent a physical threshold for pollutant concentration. A value surpassing these concentration limits would be classified as an extreme outlier and be therefore discarded.

➢ L153: "urban": While the global reanalyses are known to have difficulties to represent urban-type conditions, would it make a difference to exclude the "urban" sector from your evaluation? I wonder if particularly for NO2 and O3 (but possibly also for PM) this could still result in significantly different, and more relevant, performance statistics? (although I can see that for your future applications you exactly require knowledge about the bias correction wrt urban stations, probably).

As explained in our first answer to the first reviewer, we revised the entire study by considering now only rural, rural-regional and rural-remote background stations as the main observational dataset of our study, thus exluding urban and suburban background stations, as well as traffic and industrial point source stations. For information purpose, we also provide some brief insights on the evaluation results obtained considering only urban and suburban background stations. Further details on this realignment of the manuscript can be found in our general answer provided to the first anonymous referee.

➢ Along these lines, it could be useful if the authors include (if not now, possibly in future) a correction factor for interference in the in-situ observations of NO2 for PAN and HNO3, see also a discussion of this impact in Poraicu et al. (GMDD 2022), and references therein.

Although this approach is certainly of strong interest when evaluating NO2 fields, especially over rural areas where stronger interferences are expected given the distance to NOx emission hotspots, this would require using PAN and HNO3 fields from CAMSRA, which are likely affected by substantial uncertainties given that, to the best of our knowledge, no previous studies have performed a comprehensive evaluation of these specific chemical compounds. Therefore, in the present study, we have decided to ignore this potential limitation.

➢ Fig. 1: There is a considerable change in the observational network over time. Do you have any indications to what extent this has impact on the computed trends? Furthermore, a suspicious peak in the number of observations appear around the end of 2012 for various compounds. Do you have any understanding what has caused this?

As mentioned by the reviewer, the number of stations is not constant through time, and thus the computed trends should not be taken as a reliable estimate of the trends affecting the main pollutants over Europe. Such an analysis would require to select only stations with a sufficient amount of observations throughout the entire time period. Nevertheless, the trends are of importance for the evaluation given their relevance in the comparison between reanalysis and observations. The following sentence was introduced in the text:

"It is worth noting that these trends are here calculated essentially for evaluating the consistency of the reanalyses against observational data, but should not be taken as a reliable estimate of real pollutant trends due to the number of stations not being constant, but generally increasing throughout the period of study. Moreover, even if a station has available data over the entire period, its location can also be subject to changes over time. "

Even so, both CO and SO2 present a relatively constant number of gridded cells over the period 2003-2020, while only PM2.5 shows a very large increase from around 20 in 2003 to over 200 cells in 2020. O3, NO2 and PM10 also present significant increases in the number of gridded cells over the period of study, but the amount of observational data at the beginning of the period is already large, ranging from 200 to 400 cells, so the impact related to the change in the observational network over time for these pollutants should be limited.

The few peaks visible around 2012-2013 are likely related issues arising during the transition between EEA's AIRBASE and AQ e-Reporting databases (e.g. duplicated stations during a few months). Moreover, stations extracted through GHOST only change from month to month, so it is possible that the peak occurs due to the addition of stations from AQ e-Reporting and the subsequent removal of a large number of stations from AIRBASE, which ends in 2012. These potential issues are not expected to impact too much the results given the large number of stations considered in the study.

➢ Sec. 3.1: The authors now focus on monthly mean values. It would be nice and interesting to try also other metrics (e.g. biases, and trends in summertime, daytime ozone), particularly as this is more relevant for health-related applications. But I can understand if that is out of scope of the current work.

We acknowledge this may represent a shortcoming in our work and it is certainly an interesting idea, but given the already considerable extension of our study we think that including these additional results would render our manuscript too long. Our idea for this study was also to keep the paper as concise and digestible for the readers as possible, given it already contains a lot of information. Nevertheless, seasonal statistics are included in the manuscript for all pollutants and for both rural and urban background stations.

➢ The first couple of sentences of the abstract are of course true; on the other hand to my taste this is possibly a bit over-the-top, and high-level to motivate the work that you present here, that is not needed. It would work if you are a bit more modest here.

After reviewing the sentences and discussing them among the co-authors, we find them to be an overall objective and appropriate introduction to the manuscript. Nevertheless, some minor modifications were implemented to better convey the message we intended to put forward.

➢ L6: "…they do not integrate surface measurements…" :I'd suggest to include the word 'generally': it is true that current, specified global reanalyses do not include surface obs, but it is not a rule, I'd say.

The following modification was introduced in the text:

L6: "...they do not integrate surface measurements…" ➔ "...they generally do not integrate surface measurements…"

➢ L71-72: I understand it's easier to simply talk about concentrations, but when reporting O3 in units ppbv this is really 'volume mixing ratio' - some further clarification at this point might be good.

The persistent use of the term "concentration" throughout the text, even in the cases where we refer to a "mixing ratio", was merely for the sake of simplicity. The following modification has been introduced in the text:

L71: "Throughout this work, square brackets, [], are used to indicate concentration of a chemical compound (e.g. [O3] = O3 concentration)... ➔ "Throughout this work, square brackets, [], are used to indicate concentration or mixing ratio of a chemical compound (e.g. [O3] = O3 mixing ratio, [PM10] = PM10 concentration)...
Nonetheless, the term concentration is used for the sake of simplicity when reactive gases are mentioned together with aerosols."

The term "concentration" has been replaced for "mixing ratio" for all reactive gases.

➢ L94: "The bias…" I think it is more accurate to write something like: "The biases present in the different AC satellite retrieval datasets…"

The sentence was rewritten as follows:

L94: "The bias present in the different atmospheric composition datasets…" ➔ "The biases present in the different satellite-retrieved AC datasets…"

➢ L97: "Separate chemical compounds" better write "separate aerosol compounds"?

The sentence was rewritten as follows:

L97: "...its separate chemical compounds, which…" ➔ "...its separate aerosol compounds, which…"

➢ Table 2, "Assimilation system: IFS Cycle 42r1 4D-Var ; Meteorology: ERA5" better write here: "Assimilation system: 4D-Var ; Meteorology: IFS Cycle 42r1"

The changes were implemented as suggested.

**Referee #3**

A series of specific comments and questions are hereunder adressed by the authors:

➢ I wonder how their surface observation data (AIRBASE and AQ_eReporting from EEA) are different from observational data that were used in other previous studies. That is, did the authors use the same observation data, or are there any differences in surface observation data? For example, they said that Wagner et al. (2021) used EMEP observations. More detailed and specific description of how their data are different or the same would be helpful for readers.

AIRBASE and AQ e-Reporting are two EEA databases which gather surface observational data from the EEA's network of ground stations. There is no intrinsic difference between our observational data and data obtained from other networks. Its relevance relies on the fact that it comes from a much extensive network which includes a significantly greater number of background stations and it is filtered through several quality-assurance (QA) flags, which are included in Appendix D.

➢ It would be good and interesting to compare the performance of the two reanalyses for different settings, e.g., urban or highly polluted regions vs. remote regions, and so on.

This has been partly achieved by the separation of the overall set of background stations into two different subsets of urban and rural background stations, respectively.
The authors acknowledge this would certainly be an interesting analysis to perform, though we fear it would render the current manuscript to be too extensive, given its current length. Nevertheless, it is certainly a very interesting idea that could be studied in subsequent publications.

➢ The authors provided spatial maps of pollutants from CAMSRA and MERRA-2. How about showing the same thing (spatial map) for the surface observations? I understand that they gridded the surface observation data, and there might be missing grids because of a lack of observation data. But, spatial maps from observations would provide an intuitive insight into the atmospheric composition over Europe.

The spatial coverage of the observational network changes through time and there are many data gaps, both in space and time, which would render such a map as unrepresentative of the actual observed concentrations over the European continent.

➢ Fig. 5: I doubt if the authors correctly presented the results for SO2 in Fig. 5. In Fig. 5a, MERRA-2 SO2 concentration (in blue) is significantly higher than observed (black) and CAMSRA (green) SO2 during the study period. However, SO2 in Fig. 5c, which represents the spatial averaged SO2 from CAMSRA is apparently higher than that from MERRA-2 (Fig. 5d). I strongly suggest that the authors should double check if all the data and analysis are correct.

This apparently strange behaviour can be easily explained by considering the number and

distribution of stations across the different countries examined. The time series displayed in a) for each of the pollutants represents the spatially-averaged (medianized across all gridded cells with available observations) concentration, thus regions with a higher station density will contribute more towards the final concentration value. By examining Fig. 5c and Fig. 5d it would appear that CAMSRA should display a higher concentration than MERRA-2 in Fig. 5a, but this does not occur due to the distribution of ground stations across Europe. From Fig. 5e it can be immediately seen that MERRA-2 presents higher SO2 concentrations in several countries: Germany, France, Belgium, Denmark, Netherlands, Italy, Austria, Switzerland, Czechia, Greece, Albania, Finland, Sweden and significant areas of the UK, Ireland and Poland. In the rest of countries not mentioned above, differences between MERRA-2 and CAMSRA remain generally small, with notable exceptions localized in the Balkans and in certain regions of the Iberian Peninsula.

The following information has been added to the manuscript:

"On a first examination of the SO2 spatial distribution, it may appear as if the concentration values in the time series should be larger for CAMSRA, though this is actually misleading as the evaluation is performed only in cells with available observations. Therefore, regions with a higher station density contribute more towards the final concentration value. From Fig. 5e it can be immediately seen that MERRA-2 presents higher SO2 concentrations in several countries which have an overall larger number of stations (e.g. Germany, Netherlands, France, Italy)."

Moreover, differences between MERRA-2 and CAMSRA in their time series are less notorious if the spatial average across stations is performed using the mean instead of the median.

➢ L304: Why does MERRA-2 show a large increase in CO concentration in 2020?

After plotting the CO concentration annual means over Europe for 2019 and 2020, the large increase in CO in MERRA-2 can be attributed to an important concentration surge in several western European countries (e.g. Germany, Italy, France, Low Countries, United Kingdom).
The following sentence has been added to the manuscript:

L312: "The large increase in CO concentration observed in MERRA-2 for 2020 can be attributed to significant concentration increase between 2019 and 2020 in certain regions of the European continent (e.g. Rhine-Rühr valley, Paris/London metropolitan areas, Po River basin). This CO concentration surge can already be seen in the raw version (i.e. non-regridded) of the reanalysis."

➢ Some figures (e.g., Fig. 1, Fig. 6, Fig. 7) do not show long-term trend lines for MERRA-2.

The trend is only shown in the figures when it is statistically significant at a 99 % confidence level. This is mentioned in the caption of Fig. 2, which is essentially equal for Fig. 2-7 as the information displayed remains the same but for different pollutants.

➢ L35: Remove "to"

The modification was introduced in the text as suggested.

➢ L38: Add "and" before health studies.

The modification was introduced in the text as suggested.

➢ L74: Data comes -> data come, or revise the first several words.

The modification was introduced in the text as suggested.

➢ L146: What is "BSC"?

BSC is the acronym for Barcelona Supercomputing Center, the research institution in which the main authors are affiliated.